# Towards Principled Unsupervised Multi-Agent Reinforcement Learning

**Riccardo Zamboni**
Politecnico di Milano
riccardo.zamboni@polimi.it

**Mirco Mutti**
Technion

**Marcello Restelli**
Politecnico di Milano

## Abstract

In reinforcement learning, we typically refer to *unsupervised* pre-training when we aim to pre-train a policy without a priori access to the task specification, i.e., rewards, to be later employed for efficient learning of downstream tasks. In single-agent settings, the problem has been extensively studied and mostly understood. A popular approach casts the unsupervised objective as maximizing the *entropy* of the state distribution induced by the agent's policy, from which principles and methods follow. In contrast, little is known about state entropy maximization in multi-agent settings, which are ubiquitous in the real world. What are the pros and cons of alternative problem formulations in this setting? How hard is the problem in theory, how can we solve it in practice? In this paper, we address these questions by first characterizing those alternative formulations and highlighting how the problem, even when tractable in theory, is non-trivial in practice. Then, we present a scalable, decentralized, trust-region policy search algorithm to address the problem in practical settings. Finally, we provide numerical validations to both corroborate the theoretical findings and pave the way for unsupervised multi-agent reinforcement learning via state entropy maximization in challenging domains, showing that optimizing for a specific objective, namely *mixture entropy*, provides an excellent trade-off between tractability and performances.

## 1 Introduction

Multi-Agent Reinforcement Learning [MARL, Albrecht et al., 2024] recently showed promising results in learning complex behaviors, such as coordination and teamwork [Samvelyan et al., 2019], strategic planning in the presence of imperfect knowledge [Perolat et al., 2022], and trading [Johanson et al., 2022]. Just like in single-agent RL, however, most of the efforts are focused on tabula rasa learning, that is, without exploiting any prior knowledge gathered from offline data and/or policy pre-training. Despite its generality, learning tabula rasa hinders MARL from addressing real-world situations, where training from scratch is slow, expensive, and arguably unnecessary [Agarwal et al., 2022]. In this regard, some progress has been made on techniques specific to the multi-agent setting, ranging from ad hoc teamwork [Mirsky et al., 2022] to zero-shot coordination [Hu et al., 2020], but our understanding of what can be done *instead of* learning tabula rasa is still limited.

In single-agent RL, unsupervised pre-training frameworks [Laskin et al., 2021] have emerged as a viable solution: a policy is pre-trained without a priori access to the task specification, i.e., rewards, to be later employed for efficient learning of downstream tasks. Among others, state-entropy maximization [Hazan et al., 2019, Lee et al., 2019] was shown to be a useful tool for policy pre-training [Hazan et al., 2019, Mutti et al., 2021] and data collection for offline learning [Yarats et al., 2022]. In this setting, the unsupervised objective is cast as maximizing the entropy of the state distribution induced by the agent's policy. Recently, the potential of entropy objectives in MARL was empirically corroborated by a plethora of works [Liu et al., 2021, Zhang et al., 2021b, Yang et al., 2021,

Xu et al., 2024] investigating entropic reward-shaping techniques to boost exploration in downstream tasks. Yet, to the best of our knowledge, the literature still lacks a principled understanding of how state entropy maximization works in multi-agent settings, and how it can be used for unsupervised pre-training. Let us think of an illustrative example that highlights the central question of this work: multiple autonomous robots deployed in a factory for a production task. The robots' main goal is to perform many operations over a large set of products, with objectives ranging from optimizing for costs and energy to throughput, which may change over time depending on the market's condition. Arguably, trying to learn each possible task from scratch is inefficient and unnecessary. On the other hand, one could think of first learning to cover the possible states of the system and then fine-tune this general policy over a specific task. Yet, if everyone is focused on their own exploration, any incentive to collaborate with each other may disappear, especially when coordinating comes at a cost for individuals. Similarly, covering the entire space might be unreasonable in most real-world cases. Here we are looking for a third alternative.

> **Research Questions:**
>
> **(Q1)** Can we formulate a multi-agent counterpart of the unsupervised pre-training via state entropy maximization in a principled way?
>
> **(Q2)** How are different formulations related? Do crucial theoretical differences emerge?
>
> **(Q3)** Can we explicitly pre-train a policy for state entropy maximization in practical multi-agent scenarios?
>
> **(Q4)** Do crucial differences emerge in practice? Does this have an impact on downstream tasks learning?

**Content Outline and Contributions.** First, in Section 3, we address **(Q1)** by showing that the problem can be addressed through the lenses of a specific class of decision making problems, called *convex Markov Games* [Gemp et al., 2024, Kalogiannis et al., 2025], yet it can take different, alternative, formulations. Specifically, they differ on whether the agents are trying to *jointly* cover the space through conditionally dependent actions, or they neglect the presence of others and deploy fully *disjoint* strategies, or they coordinate to cover the state space beforehand, but taking actions independently as components of a *mixture*. We formalize these cases into three distinct objectives. Then, in Section 4, we address **(Q2)**, highlighting that these objectives are related through performance bounds that scale with the number of agents. We also show that only the joint or mixture objectives enjoy remarkable convergence properties under policy gradient updates in the ideal case of evaluating the agents' performance over infinite realizations (trials). However, as one shifts the attention to the more practical scenario of reaching good performance over a handful, or even just one, trial, we show that different objectives lead to different behaviors and mixture objectives do enjoy more favorable properties. Then, in Section 5, we address **(Q3)** by introducing a decentralized multi-agent policy optimization algorithm, called *Trust Region Pure Exploration* (TRPE), explicitly addressing state entropy maximization pre-training over finite trials. Finally, we address **(Q4)** by testing the algorithm on some simple yet challenging settings, showing that optimizing for a specific objective, namely mixture entropy, provides an excellent trade-off between tractability and performances. We show that this objective yields superior sample complexity and remarkable zero-shot performance when the pre-trained policy is deployed in sparse reward downstream tasks.

## 2 Preliminaries

In this section, we introduce the most relevant background and the basic notation.

**Notation.** We denote $[N] := \{1, 2, \ldots, N\}$ for a constant $N < \infty$. We denote a set with a calligraphic letter $\mathcal{A}$ and its size as $|\mathcal{A}|$. For a (finite) set $\mathcal{A} = \{1, 2, \ldots, i, \ldots\}$, we denote $-i = \mathcal{A}/\{i\}$ the set of all its elements but $i$. $\mathcal{A}^T := \times_{t=1}^{T} \mathcal{A}$ is the $T$-fold Cartesian product of $\mathcal{A}$. The simplex on $\mathcal{A}$ is $\Delta_{\mathcal{A}} := \{p \in [0, 1]^{|\mathcal{A}|} | \sum_{a \in \mathcal{A}} p(a) = 1\}$ and $\Delta_{\mathcal{A}}^{\mathcal{B}}$ denotes the set of conditional distributions $p : \mathcal{A} \to \Delta_{\mathcal{B}}$. Let $X, X'$ random variables on the set of outcomes $\mathcal{X}$ and corresponding probability measures $p_X, p_{X'}$, we denote the Shannon entropy of $X$ as $H(X) = -\sum_{x \in \mathcal{X}} p_X(x) \log(p_X(x))$ and the Kullback-Leibler (KL) divergence as $D_{\mathrm{KL}}(p_X \| p_{X'}) = \sum_{x \in \mathcal{X}} p_X(x) \log(p_X(x)/p_{X'}(x))$. We denote $\mathbf{x} = (X_1, \ldots, X_T)$ a random vector of size $T$ and $\mathbf{x}[t]$ its entry at position $t \in [T]$.

**Interaction Protocol.** As a model for interaction, we consider finite-horizon Markov Games [MGs, Littman, 1994] *without rewards*. A MG $\mathcal{M} := (\mathcal{N}, \mathcal{S}, \mathcal{A}, \mathbb{P}, \mu, T)$ is composed of a set of agents $\mathcal{N}$, a set $\mathcal{S} = \times_{i\in[\mathcal{N}]}\mathcal{S}_i$ of states, and a set of (joint) actions $\mathcal{A} = \times_{i\in[\mathcal{N}]}\mathcal{A}_i$, which we assume to be discrete and finite in size $|\mathcal{S}|, |\mathcal{A}|$ respectively. At the start of an episode, the initial state $s_1$ of $\mathcal{M}$ is drawn from an initial state distribution $\mu \in \Delta_\mathcal{S}$. Upon observing $s_1$, each agent takes action $a_1^i \in \mathcal{A}_i$, the system transitions to $s_2 \sim \mathbb{P}(\cdot|s_1, a_1)$ according to the *transition model* $\mathbb{P} \in \Delta_{\mathcal{S}\times\mathcal{A}}^\mathcal{S}$. The process is repeated until $s_T$ is reached and $s_T$ is generated, being $T < \infty$ the horizon of an episode. Each agent acts according to a *policy*, that can be either Markovian when the action is only conditioned on the current state, i.e., $\pi^i \in \Delta_\mathcal{S}^{\mathcal{A}^i}$, or non-Markovian when the action is conditioned on the history, i.e., $\pi^i \in \Delta_{\mathcal{S}^t \times \mathcal{A}^t}^{\mathcal{A}^i}$.[1] Also, we will denote as *decentralized-information* policies the ones conditioned on either $\mathcal{S}_i$ or $\mathcal{S}_i^t \times \mathcal{A}_i^t$ for agent $i$, and *centralized-information* ones the ones conditioned over the full state or state-actions sequences. It follows that the joint action is taken according to the *joint* policy $\Delta_\mathcal{S}^\mathcal{A} \ni \pi = (\pi^i)_{i\in[\mathcal{N}]}$.

**Induced Distributions.** Now, let us denote as $S$ and $S_i$ the random variables corresponding to the joint state and $i$-th agent state respectively. Then the former is distributed as $d^\pi \in \Delta_\mathcal{S}$, where $d^\pi(s) = \frac{1}{T}\sum_{t\in[T]} Pr(s_t = s|\pi, \mu)$, the latter is distributed as $d_i^\pi \in \Delta_{\mathcal{S}_i}$, where $d_i^\pi(s_i) = \frac{1}{T}\sum_{t\in[T]} Pr(s_{t,i} = s_i|\pi, \mu)$. Furthermore, let us denote with $\mathbf{s}, \mathbf{a}$ the random vectors corresponding to sequences of (joint) states, and actions of length $T$, which are supported in $\mathcal{S}^T, \mathcal{A}^T$ respectively. We define $p^\pi \in \Delta_{\mathcal{S}^T \times \mathcal{A}^T}$, where $p^\pi(\mathbf{s}, \mathbf{a}) = \prod_{t\in[T]} Pr(s_t = \mathbf{s}[t], a_t = \mathbf{a}[t])$. Finally, we denote the empirical state distribution induced by $K \in \mathbb{N}^+$ trajectories $\{\mathbf{s}_k\}_{k\in[K]}$ as $d_K(s) = \frac{1}{KT}\sum_{k\in[K]}\sum_{t\in[T]} \mathbb{1}(\mathbf{s}_k[t] = s)$.

**Convex MDPs and State Entropy Maximization.** In the MDP setting ($|\mathcal{N}| = 1$), the problem of state entropy maximization can be viewed as a special case of *convex RL* [Hazan et al., 2019, Zhang et al., 2020, Zahavy et al., 2021]. In such framework, the general task is defined via an F-bounded concave[2] utility function $\mathcal{F} : \Delta_\mathcal{S} \to (-\infty, F]$, with $F < \infty$, that is a function of the state distribution $d^\pi$. This allows for a generalization of the standard RL objective, which is a linear product between a reward vector and the state(-action) distribution [Puterman, 2014]. Usually, some regularity assumptions are enforced on the function $\mathcal{F}$. In the following, we align with the literature through the following smoothness assumption:

**Assumption 2.1** (Lipschitz). *A function* $\mathcal{F} : \mathcal{A} \to \mathbb{R}$ *is Lipschitz-continuous for some constant* $L < \infty$, *or L-Lipschitz for short, if it holds* $|\mathcal{F}(x) - \mathcal{F}(y)| \leqslant L\|x - y\|_1, \ \forall(x, y) \in \mathcal{A}^2$.

More recently, Mutti et al. [2022a] noticed that in many practical scenarios only a finite number of $K \in \mathbb{N}^+$ episodes/trials can be drawn while interacting with the environment, and in such cases one should focus on $d_K$ rather than $d^\pi$. As a result, they contrast the *infinite-trials* objective defined as $\zeta_\infty(\pi) := \mathcal{F}(d^\pi)$ with a *finite-trials* one, namely $\zeta_K(\pi) := \mathbb{E}_{d_K \sim p_K^\pi} \mathcal{F}(d_K)$, noticing that convex MDPs (cMDPs) are characterized by the fact that $\zeta_K(\pi) \leqslant \zeta_\infty(\pi)$, differently from standard (linear) MDPs for which equality holds. In single-agent convex RL, state entropy maximization is defined as solving a cMDP equipped with an entropy functional [Hazan et al., 2019], namely $\mathcal{F}(d^\pi) := H(d^\pi)$.

Interestingly, even in single-agent settings, the infinite-trials state entropy objective can be formulated as a non-Markovian reward, as the *value* of being in a state depends on the states visited before and after that state.[3] As a consequence, it is not possible to derive Bellman operators of any kind [Takács, 1966, Whitehead and Lin, 1995, Zhang et al., 2020]. Conversely, for finite-trials formulations, it is possible to define a Bellman operator by extending the state representation to include the whole trajectories of interaction. Unfortunately though, even this option is intractable as the size of such an extended MDP will grow exponentially.[4]

# 3  Problem Formulation

This section addresses the first of the research questions outlined in the introduction.

---

[1] In general, we will denote the set of valid per-agent policies with $\Pi^i$ and the set of joint policies with $\Pi$.

[2] In practice, the function can be either convex, concave, or even non-convex. The term is used to distinguish the objective from the standard (linear) RL objective. We will assume $\mathcal{F}$ is concave if not mentioned otherwise.

[3] By conditioning with respect to the policy, such a reward would result to be Markovian. However, the contraction argument does not appear to hold for a Bellman operator over this kind of policy-based rewards.

[4] Indeed, the optimization of the finite-trial formulation is NP-hard [Mutti et al., 2023].

**(Q1)** *Can we formulate a multi-agent counterpart of the unsupervised pre-training via state entropy maximization in a principled way?*

In fact, when a reward function is not available, the core of the problem resides in finding a well-behaved problem formulation coherent with the task. Gemp et al. [2024] recently introduced a convex generalization of MGs called **convex Markov Games** (cMGs), namely a tuple $\mathcal{M}_\mathcal{F} := (\mathcal{N}, \mathcal{S}, \mathcal{A}, \mathbb{P}, \mathcal{F}, \mu, T)$, that consists in a MG equipped with (non-linear) functions of the *stationary joint state* distribution $\mathcal{F}(d^\pi)$. We expand over this definition, by noticing that state entropy maximization can be casted as solving a cMG equipped with an entropy functional, namely $\mathcal{F}(\cdot) := H(\cdot)$. Yet, important new questions arise: Over which distributions should agents compute the entropy? How much information should they have access to? Can we define objectives accounting for a finite number of trials? Different answers depict different objectives.

**Joint Objectives.** The first and most straightforward way to formulate the problem is to define it as in the MDP setting, with the joint state distribution simply taking the place of the single-agent state distribution. In this case, we define *infinite-trials* and *finite-trials Joint* objectives, respectively

$$\max_{\pi=(\pi^i \in \Pi^i)_{i \in [\mathcal{N}]}} \left\{ \zeta_\infty(\pi) := \mathcal{F}(d^\pi) \right\} \qquad \max_{\pi=(\pi^i \in \Pi^i)_{i \in [\mathcal{N}]}} \left\{ \zeta_K(\pi) := \mathbb{E}_{d_K \sim p_K^\pi} \mathcal{F}(d_K) \right\} \quad (1)$$

In state entropy maximization tasks, an optimal (joint) policy will try to cover the joint state space uniformly, either in expectation or over a finite number of trials respectively. In this, the joint formulation is rather intuitive as it describes the most general case of multi-agent exploration. Moreover, as each agent sees a difference in performance explicitly linked to others, this objective should be able to foster coordinated exploration. As we shall see, this comes at a price.

**Disjoint Objectives.** One might look for formulations that fully embrace the multi-agent setting, such as defining a set of functions supported on per-agent state distributions rather than joint distributions. This intuition leads to *infinite-trials* and *finite-trials Disjoint* objectives:

$$\left\{ \max_{\pi^i \in \Pi^i} \zeta_\infty^i(\pi^i, \cdot) := \mathcal{F}(d_i^{\pi^i, \cdot}) \right\}_{i \in [\mathcal{N}]} \qquad \left\{ \max_{\pi^i \in \Pi^i} \zeta_K^i(\pi^i, \cdot) := \mathbb{E}_{d_K \sim p_K^{\pi^i, \cdot}} \mathcal{F}(d_{K,i}) \right\}_{i \in [\mathcal{N}]} \quad (2)$$

According to these objectives, each agent will try to maximize her own marginal state entropy separately, neglecting the effect of her actions over others performances. In other words, we expect this objective to hinder the potential coordinated exploration, where one has to take as step down as so allow a better performance overall.

**Mixture Objectives.** At last, we introduce a problem formulation that will later prove capable of uniquely taking advantage of the structure of the problem. First, we introduce the following:

**Assumption 3.1** (Uniformity). *The agents have the same state space* $\mathcal{S}_i = \mathcal{S}_j = \tilde{\mathcal{S}}, \forall (i,j) \in \mathcal{N} \times \mathcal{N}$.[5]

Under this assumption, we will drop the agent subscript when referring to the per-agent states and use $\tilde{\mathcal{S}}$ instead. Interestingly, this assumption allows us to define a particular distribution:

$$\tilde{d}^\pi(\tilde{s}) := \frac{1}{|\mathcal{N}|} \sum_{i \in [\mathcal{N}]} d_i^\pi(\tilde{s}) \in \Delta_{\tilde{\mathcal{S}}}.$$

We refer to this distribution as *mixture* distribution, given that it is defined as a uniform mixture of the per-agent marginal distributions. Intuitively, it describes the average probability over all the agents to be in a common state $\tilde{s} \in \tilde{\mathcal{S}}$, in contrast with the joint

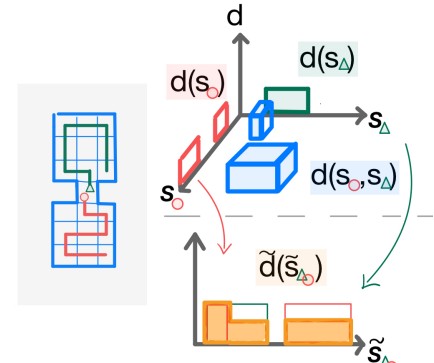

Figure 1: The interaction on the *left* induces different (empirical) distributions: Marginal distributions for **agent 1** and **agent 2** over their respective states; a **joint distribution** over the product space; a **mixture distribution** over a common space, defined as the average. The mixture distribution is usually *less sparse*.

---

[5]One should notice that even in cMGs where this is not (even partially) the case, the assumption can be enforced by padding together the per-agent states.

distribution that describes the probability for them to be in a joint state $s$, or the marginals that describes the probability of each one of them separately. In Figure 1 we provide a visual representation of these concepts. Similarly to what happens for the joint distribution, one can define the empirical distribution induced by $K$ episodes as $\tilde{d}_K(\tilde{s}) = \frac{1}{|\mathcal{N}|} \sum_{i\in[\mathcal{N}]} d_{K,i}(\tilde{s})$ and $\tilde{d}^\pi = \mathbb{E}_{\tilde{d}_K \sim p_K^\pi}[\tilde{d}_K]$. The mixture distribution allows for the definition of the *Mixture* objectives, in their infinite and finite trials formulations respectively:

$$\max_{\pi=(\pi^i\in\Pi^i)_{i\in[\mathcal{N}]}} \left\{ \tilde{\zeta}_\infty(\pi) := \mathcal{F}(\tilde{d}^\pi) \right\} \qquad \max_{\pi=(\pi^i\in\Pi^i)_{i\in[\mathcal{N}]}} \left\{ \tilde{\zeta}_K(\pi) := \mathbb{E}_{\tilde{d}_K \sim p_K^\pi} \mathcal{F}(\tilde{d}_K) \right\} \qquad (3)$$

When this kind of objectives is employed in state entropy maximization, the entropy of the mixture distribution decomposes as $H(\tilde{d}^\pi) = \frac{1}{|\mathcal{N}|} \sum_{i\in[\mathcal{N}]} H(d_i^\pi) + \frac{1}{|\mathcal{N}|} \sum_{i\in[\mathcal{N}]} D_{\mathrm{KL}}(d_i^\pi||\tilde{d}^\pi)$ and one remarkable scenario arises: Agents follow policies possibly inducing lower disjoint entropies, but their induced marginal distributions are maximally different. Thus, the average entropy remains low, but the overall mixture entropy is high due to diversity (i.e., high values of the KL divergences). This scenario has been referred to in Kolchinsky and Tracey [2017] as the *clustering* scenario and, in the following, we will provide additional evidences why this scenario is particularly relevant.

## 4   A Formal Characterization of Multi-Agent State Entropy Maximization

In the previous section, we provided a principled problem formulation of multi-agent state entropy maximization through an array of different objectives. Here, we address the second research question:

> (**Q2**) *How are different formulations related? Do crucial theoretical differences emerge?*

First of all, we show that if we look at state entropy maximization tasks specifically, i.e. cMGs $\mathcal{M}_H$ equipped with entropy functionals $\mathcal{F}(\cdot) := H(\cdot)$, all the objectives in infinite-trials formulation can be elegantly linked one to the other though the following result:

**Lemma 4.1** (Entropy Mismatch). *For every cMG $\mathcal{M}_H$, for a fixed (joint) policy $\pi = (\pi^i)_{i\in\mathcal{N}}$ the infinite-trials objectives are ordered according to:*

$$\frac{H(d^\pi)}{|\mathcal{N}|} \leqslant \frac{1}{|\mathcal{N}|} \sum_{i\in[\mathcal{N}]} H(d_i^\pi) \leqslant H(\tilde{d}^\pi) \leqslant \sup_{i\in[\mathcal{N}]} H(d_i^\pi) + \log(|\mathcal{N}|) \leqslant H(d^\pi) + \log(|\mathcal{N}|)$$

The full derivation of these bounds is reported in Appendix B. This set of bounds demonstrates that the difference in performances over infinite-trials objective for the same policy can be generally bounded as a function of the number of agents. In particular, disjoint objectives generally provides poor approximations of the joint objective from the point of view of the single-agent, while the mixture objective is guaranteed to be a rather good lower bound to the joint entropy as well, since its over-estimation scales logarithmically with the number of agents.

It is still an open question how hard it is to actually optimize for these objectives. Now, while general cMGs $\mathcal{M}_\mathcal{F}$ are an interaction framework whose general properties are far from being well-understood, they surely enjoy some nice properties. In particular, it is possible to exploit the fact that performing Policy Gradient [PG, Sutton et al., 1999, Peters and Schaal, 2008] independently among the agents is equivalent to running PG jointly, since this is done over the same common objective as for Potential Markov Games [Leonardos et al., 2022] (see Lemma B.5 in Appendix B.1). This allows us to provide a rather positive answer, here stated informally and extensively discussed in Appendix B.1:

**Fact 4.1** ((Informal) Sufficiency of Independent Policy Gradient). *Under proper assumptions, for every cMG $\mathcal{M}_\mathcal{F}$, independent Policy Gradient over infinite trials non-disjoint objectives via centralized-information policies of the form $\pi = (\pi^i \in \Delta_\mathcal{S}^{\mathcal{A}^i})_{i\in[\mathcal{N}]}$ converges fast.*

This result suggests that PG should be generally enough for the infinite-trials optimization, and thus, in some sense, these problems might not be of so much interest. However, cMDP theory has outlined that optimizing for infinite-trials objectives might actually lead to extremely poor performance as soon as the policies are deployed over just a handful of trials, i.e. in almost any practical scenario [Mutti et al., 2023]. We show that this property transfers almost seamlessly to cMGs as well, with interesting additional take-outs:

**Theorem 4.2** (Finite-Trials Mismatch in cMGs). *For every cMG $\mathcal{M}_\mathcal{F}$ equipped with a L-Lipschitz function $\mathcal{F}$, let $K \in \mathbb{N}^+$ be a number of evaluation episodes/trials, and let $\delta \in (0,1]$ be a confidence level, then for any (joint) policy $\pi = (\pi^i \in \Pi^i)_{i \in [\mathcal{N}]}$, it holds that*

$$|\zeta_K(\pi) - \zeta_\infty(\pi)| \leqslant LT\sqrt{\frac{2|\mathcal{S}|\log(2T/\delta)}{K}}, \quad |\zeta_K^i(\pi) - \zeta_\infty^i(\pi)| \leqslant LT\sqrt{\frac{2|\tilde{\mathcal{S}}|\log(2T/\delta)}{K}},$$

$$|\tilde{\zeta}_K(\pi) - \tilde{\zeta}_\infty(\pi)| \leqslant LT\sqrt{\frac{2|\tilde{\mathcal{S}}|\log(2T/\delta)}{|\mathcal{N}|K}}.$$

In general, this set of bounds confirms that for any given policy, infinite and finite trials performances might be extremely different, and thus optimizing the infinite-trials objective might lead to unpredictable performance at deployment, whenever this is done over a handful of trials. This property is inherently linked to the *convex* nature of cMGs, and Mutti et al. [2023] introduced it for cMDPs to highlight that the concentration properties of empirical state-distributions Weissman et al. [2003] allow for a nice dependency on the number of trials in controlling the mismatch. In multi-agent settings, the result portraits a more nuanced scene:

*(i)* The mismatch still scales with the cardinality of the support of the state distribution, yet, for joint objectives, this quantity scales very poorly in the number of agents.[6] Thus, even though optimizing infinite-trials joint objectives might be rather easy *in theory* as Fact 4.1 suggests, it might result in poor performances *in practice*. On the other hand, the quantity is independent of the number of agents for disjoint and mixture objectives.

*(ii)* Looking at mixture objectives, the mismatch scales sub-linearly with the number of agents $\mathcal{N}$. In some sense, the number of agents has the same role as the number of trials: The more the agents the less the deployment mismatch, and at the limit, with $\mathcal{N} \to \infty$, the mismatch vanishes completely.[7] In other words, this result portraits a striking difference with respect to joint objectives: When facing state entropy maximization over mixtures, a reasonably high number of agents compared to the size of the state-space actually helps, and simple policy gradient over mixture objectives might be enough.

**Remark 1.** Although we do not claim that the mixture objective is the one-fits-all solution, it is nonetheless *well-founded*. In particular whenever the rewards the agents will face in downstream tasks are equivalent for every agent, as it happens in relevant practical settings. When, on the other hand, the agents will aim to visit every joint state while solving for a specific task,[8] the joint entropy objective is preferable, although it may be impractical: We report in Appendix A an overall comparison of the two options, providing a motivating example as well.

**Remark 2.** Fact 4.1 is valid for *centralized-information* policies only. Up to our knowledge, no guarantees are known for *decentralized-information* policies even in linear MGs. Interestingly though, the finite-trials formulation does offer additional insights on the behavior of optimal decentralized-information policies: The interested reader can learn more about this in Appendix B.2.

## 5 An Algorithm for Multi-Agent State Entropy Maximization in Practice

As stated before, a core drive of this work is addressing practical scenarios, where only a handful of trials can be drawn while interacting with the environment. Yet, Th. 4.2 implies that optimizing for infinite-trials objectives, as with PG updates in Fact 4.1, might result in poor performance at deployment. As a result, here we address the third research question, that is:

> **(Q3)** *Can we explicitly pre-train a policy for state entropy maximization in practical multi-agent scenarios?*

To do so, we will shift our attention from infinite trials objectives to finite trials ones explicitly, more specifically on the single-trial case with $K = 1$. Remarkably, it is possible to

---

[6]Indeed, in the case of product state-spaces $\mathcal{S} = \times_{i \in [\mathcal{N}]} \mathcal{S}_i$ the cardinality scales exponentially with the number of agents $|\mathcal{N}|$.

[7]In this scenario, all the bounds of Lemma 4.1 linking different objectives become vacuous.

[8]For instance, when for two agents the reward $r(s, s')$ is different from $r(s', s)$, i.e. the order matters.

directly optimize the single-trial objective in multi-agent cases with decentralized algorithms: We introduce *Trust Region Pure Exploration* (TRPE), the first decentralized algorithm that explicitly addresses single-trial objectives in cMGs, with state entropy maximization as a special case. TRPE takes inspiration from trust-region based methods as TRPO [Schulman et al., 2015] due to their ability to address brittle optimization landscapes in which a small change into the policy parameters of each agent may drastically change the value of the objective function and the use of the trust region, like in TRPE, allows for accounting for this effect.[9] While the TRPE algorithm is new, the benefits of trust-region methods in multi-agent settings recently enjoyed an ubiquitous success and interest for their surprising effectiveness [Yu et al., 2022].

In fact, trust-region analysis nicely align with the properties of finite-trials formulations and allow for an elegant extension to cMGs through the following.

**Definition 5.1** (Surrogate Function over a Single Trial). *For every cMG $\mathcal{M}_{\mathcal{F}}$ equipped with a L-Lipschitz function $\mathcal{F}$, let $d_1$ be a general single-trial distribution $d_1 = \{d_1, d_{1,i}, \tilde{d}_1\}$, then for any per-agent deviation over policies $\pi = (\pi^i, \pi^{-i})$, $\tilde{\pi} = (\tilde{\pi}^i, \pi^{-i})$, it is possible to define a per-agent* Surrogate Function *$\mathcal{L}^i(\tilde{\pi}/\pi)$ of the form $\mathcal{L}^i(\tilde{\pi}/\pi) = \mathbb{E}_{d_1 \sim p_1^\pi} \rho_{\tilde{\pi}/\pi}^i \mathcal{F}(d_1)$, where $\rho^i$ is the per-agent importance-weight coefficient $\rho_{\tilde{\pi}/\pi}^i = p_1^{\tilde{\pi}}/p_1^\pi = \prod_{t \in [T]} \frac{\tilde{\pi}^i(\mathbf{a}^i[t]|\mathbf{s}^i[t])}{\pi^i(\mathbf{a}^i[t]|\mathbf{s}^i[t])}$.*

From this definition, it follows that the trust-region algorithmic blueprint of Schulman et al. [2015] can be directly applied to single-trial formulations, with a parametric space of stochastic differentiable policies for each agent $\Theta = \{\pi_{\theta^i}^i : \theta^i \in \Theta^i \subseteq \mathbb{R}^q\}$.

---

**Algorithm**: Trust Region Pure Exploration (**TRPE**)

1: **Input**: exploration horizon $T$, trajectories $N$, trust-region threshold $\delta$, learning rate $\eta$
2: Initialize $\boldsymbol{\theta} = (\theta^i)_{i \in [\mathcal{N}]}$
3: **for** epoch $= 1, 2, \ldots$ until convergence **do**
4:     Collect $N$ trajectories with $\pi_{\boldsymbol{\theta}} = (\pi_{\theta^i}^i)_{i \in [\mathcal{N}]}$
5:     **for** agent $i = 1, 2, \ldots$ **concurrently do**
6:         Set datasets $\mathcal{D}^i = \{(\mathbf{s}_n^i, \mathbf{a}_n^i), \zeta_1^n\}_{n \in [N]}$
7:         $h = 0, \theta_h^i = \theta^i$
8:         **while** $D_{\mathrm{KL}}(\pi_{\theta_h^i}^i \| \pi_{\theta_0^i}^i) \leqslant \delta$ **do**
9:             Compute $\hat{\mathcal{L}}^i(\theta_h^i/\theta_0^i)$ via IS as in Eq. (4)
10:            $\theta_{h+1}^i = \theta_h^i + \eta \nabla_{\theta_h^i} \hat{\mathcal{L}}^i(\theta_h^i/\theta_0^i)$
11:            $h \leftarrow h + 1$
12:         **end while**
13:         $\theta^i \leftarrow \theta_h^i$
14:     **end for**
15: **end for**
16: **Output**: joint policy $\pi_{\boldsymbol{\theta}} = (\pi_{\theta^i}^i)_{i \in [\mathcal{N}]}$

---

In practice, KL-divergence is employed for greater scalability provided a trust-region threshold $\delta$, we address the following optimization problem for each agent:

$$\max_{\tilde{\theta}^i \in \Theta^i} \mathcal{L}^i(\tilde{\theta}^i/\theta^i) \qquad \text{s.t. } D_{\mathrm{KL}}(\pi_{\tilde{\theta}^i}^i \| \pi_{\theta^i}^i) \leqslant \delta$$

where we simplified the notation by letting $\mathcal{L}^i(\tilde{\theta}^i/\theta^i) := \mathcal{L}^i(\pi_{\tilde{\theta}^i}^i, \pi_{\theta^{-i}}^{-i}/\pi_\theta)$.[10]

The main idea then follows from noticing that the surrogate function in Def. 5.1 consists of an Importance Sampling (IS) estimator [Owen, 2013], and it is then possible to optimize it in a fully decentralized and off-policy manner [Metelli et al., 2020, Mutti and Restelli, 2020]. More specifically, given a pre-specified objective of interest $\zeta_1 \in \{\zeta_1, \zeta_1^i, \tilde{\zeta}_1\}$, agents sample $N$ trajectories $\{(\mathbf{s}_n, \mathbf{a}_n)\}_{n \in [N]}$ following a (joint) policy with parameters $\boldsymbol{\theta}_0 = (\theta_0^i, \theta_0^{-i})$. They then compute the values of the objective for each trajectory, building separate datasets $\mathcal{D}^i = \{(\mathbf{s}_n^i, \mathbf{a}_n^i), \zeta_1^n\}_{n \in [N]}$ and using it to compute the Monte-Carlo approximation of the surrogate function, namely

$$\hat{\mathcal{L}}^i(\theta_h^i/\theta_0^i) = \frac{1}{N} \sum_{n \in [N]} \rho_{\theta_h^i/\theta_0^i}^{i,n} \zeta_1^n, \quad \rho_{\theta_h^i/\theta_0^i}^{i,n} = \prod_{t \in [T]} \pi_{\theta_h^i}^i(\mathbf{a}_n^i[t]|\mathbf{s}_n^i[t])/\pi_{\theta_0^i}^i(\mathbf{a}_n^i[t]|\mathbf{s}_n^i[t]), \qquad (4)$$

and $\zeta_1^n$ is the plug-in estimator of the entropy based on the empirical measure $d_1$ [Paninski, 2003]. Finally, at each off-policy iteration $h$, each agent updates its parameter via gradient ascent $\theta_{h+1}^i \leftarrow \theta_h^i + \eta \nabla_{\theta_h^i} \hat{\mathcal{L}}^i(\theta_h^i/\theta_0^i)$ until the trust-region boundary is reached, i.e., when it holds $D_{\mathrm{KL}}(\pi_{\tilde{\theta}^i}^i \| \pi_{\theta^i}^i) > \delta$. The psudo-code of TRPE is reported in Algorithm 1. We remark that even though TRPE is applied to

---

[9] Previous works have connected the trust region with the natural gradient [Pajarinen et al., 2019].

[10] More precisely, $\mathcal{L}^i(\pi_{\tilde{\theta}^i}^i, \pi_{\theta^{-i}}^{-i}/\pi_\theta) = \mathbb{E}_{d_1 \sim p_1^{\pi_\theta}} p_1^{\pi_{\tilde{\theta}^i}^i, \pi_{\theta^{-i}}^{-i}}/p_1^{\pi_{\theta^i}^i, \pi_{\theta^{-i}}^{-i}} \mathcal{F}(d_1)$.

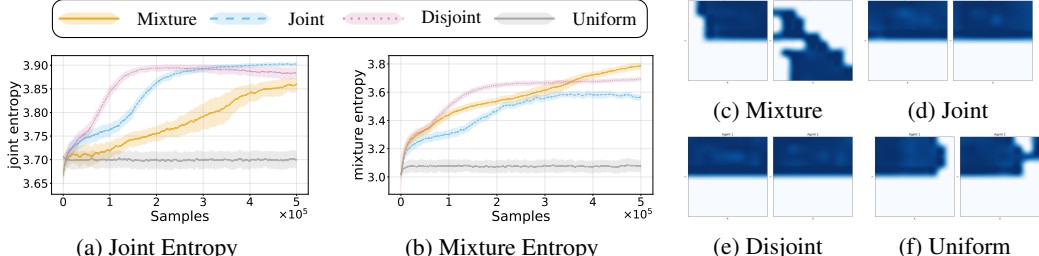

Figure 2: Single-trial Joint and Mixture Entropy induced by different objective optimization along a $T = 50$ horizon. (*Right*) State Distributions of two agents induced by different learned policies. We report the average and 95% confidence interval over 4 runs.

state entropy maximization, the algorithmic blueprint does not explicitly require the function $\mathcal{F}$ to be the entropy function and thus it is of independent interest.

**Limitations.** The main limitations of the proposed methods are two. First, the Monte-Carlo estimation of single-trial objectives might be sample-inefficient in high-dimensional tasks. However, more efficient estimators of single-trial objectives remain an open question in single-agent convex RL as well, as the convex nature of the problem hinders the applicability of Bellman operators. Secondly, the plug-in estimator of the entropy is applicable to discrete spaces only, but designing scalable estimators of the entropy in continuous domains is usually a contribution *per se* [Mutti et al., 2021].

## 6 Empirical Corroboration

In this section, we address the last research question, that is:

> (**Q4**) *Do crucial differences emerge in practice? Does this have an impact on downstream tasks learning?*

by providing empirical corroboration of the findings discussed so far. Especially, we aim to answer the following questions: (**Q4.1**) Is Algorithm 1 actually capable of optimizing finite-trials objectives? (**Q4.2**) Do different objectives enforce different behaviors, as expected from Section 3? (**Q4.3**) Does the *clustering* behavior of mixture objectives play a crucial role? If yes, when and why?

Throughout the experiments, we will compare the result of optimizing finite-trial objectives, either joint, disjoint, mixture ones, through Algorithm 1 via fully decentralized policies. The experiments will be performed with different values of the exploration horizon $T$, so as to test their capabilities in different exploration efficiency regimes.[11] The full implementation details are reported in Appendix C.

**Experimental Domains.** The experiments were performed with the aim to illustrate essential features of state entropy maximization suggested by the theoretical analysis, and for this reason the domains were selected for being challenging while keeping high interpretability. The first is a notoriously difficult multi-agent exploration task called *secret room* [MPE, Liu et al., 2021],[12] referred to as Env. (**i**). In such task, two agents are required to reach a target while navigating over two rooms divided by a door. In order to keep the door open, at least one agent have to remain on a switch. Two switches are located at the corners of the two rooms. The hardness of the task then comes from the need of coordinated exploration, where one agent allows for the exploration of the other. The second is a simpler exploration task yet over a high dimensional state-space, namely a 2-agent instantiation of *Reacher* [MaMuJoCo, Peng et al., 2021], referred to as Env. (**ii**). Each agent corresponds to one joint and equipped with decentralized-information policies. In order to allow for the use of plug-in estimator of the entropy [Paninski, 2003], each state dimension was discretized over 10 bins.

**State Entropy Maximization.** As common for the unsupervised RL framework [Hazan et al., 2019, Laskin et al., 2021, Liu and Abbeel, 2021b, Mutti et al., 2021], Algorithm 1 was first tested in her

---

[11]The exploration horizon $T$, rather than being a given trajectory length, has to be seen as a parameter of the exploration phase which allows to tradeoff exploration quality with exploration efficiency.

[12]We highlight that all previous efforts in this task employed centralized-information policies. On the other hand, we are interested on the role of the entropic feedback in fostering coordination rather than full-state conditioning, thus we employed decentralized-information policies.

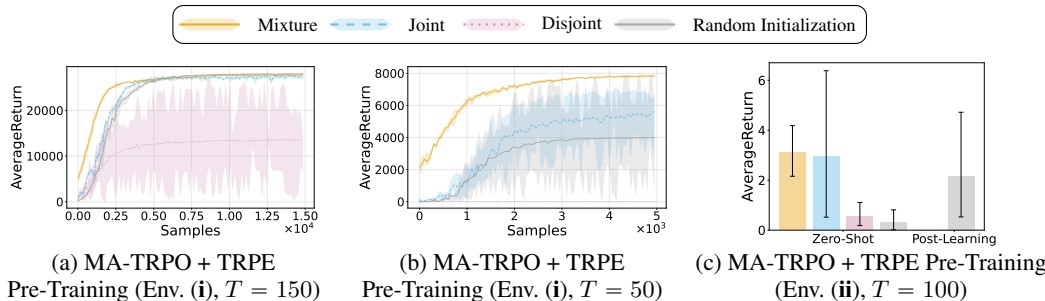

|  (a) MA-TRPO + TRPE
Pre-Training (Env. (**i**), $T = 150$) | (b) MA-TRPO + TRPE
Pre-Training (Env. (**i**), $T = 50$) | (c) MA-TRPO + TRPE Pre-Training
(Env. (**ii**), $T = 100$) |

Figure 3: Effect of pre-training in sparse-reward settings. (*Left*) Policies initialized with either Uniform or TRPE pre-trained policies. (*Right*) Policies initialized with either Zero-Mean or TRPE pre-trained policies. We report the average and 95% c.i. over 4 runs over worst-case goals.

ability to optimize for state entropy maximization objectives, thus in environments *without rewards*. In Figure 2, we report the results for a short, and thus more challenging, exploration horizon ($T = 50$) over Env. (**i**), as it is far more interpretable. Other experiments with longer horizons or over Env. (**ii**) can be found in Appendix C. Interestingly, at this challenging exploration regime, when looking at the joint entropy in Figure 2a, joint and disjoint objectives perform rather well compared to mixture ones in terms of induced joint entropy, while they fail to address mixture entropy explicitly, as seen in Figure 2b. On the other hand mixture-based objectives result in optimizing both mixture *and* joint entropy effectively, as one would expect by the bounds in Th. 4.1. By looking at the actual state visitation induced by the trained policies, the difference between the objectives is apparent. While optimizing joint objectives, agents exploit the high-dimensionality of the joint space to induce highly entropic distributions even without exploring the space uniformly via coordination (Fig. 2d); the same outcome happens in disjoint objectives, with which agents focus on over-optimizing over a restricted space loosing any incentive for coordinated exploration (Fig. 2e). On the other hand, mixture objectives enforce a clustering behavior (Fig. 2c) and result in a better efficient exploration.[13]

**Policy Pre-Training via State Entropy Maximization.** Importantly, while metrics in Fig. 2 are indeed interesting qualitative metrics, especially to understand how the unsupervised optimization process works, they do not fully capture the ultimate goal in a vacuum: the ultimate goal of unsupervised (MA)RL is to provide good pre-trained models for (MA)RL. As such, the most important experimental metric to look at is the return achieve in downstream tasks, where the policy optimizing the mixture entropy fares well in comparison to others. Thus, we tested the effect of pre-training policies via state entropy maximization as a way to alleviate the well-known hardness of *sparse-reward* settings. In order to do so, we employed a multi-agent counterpart of the TRPO algorithm Schulman et al. [2015] with different pre-trained policies. First, we investigated the effect on the learning curve in the hard-exploration task of Env. (**i**) under long horizons ($T = 150$), with a *worst-case goal* set on the opposite corner of the closed room. Pre-training via mixture objectives still lead to a faster learning compared to initializing the policy with a uniform distribution. On the other hand, joint objective pre-training did not lead to substantial improvements over standard initializations. More interestingly, when extremely short horizons were taken into account ($T = 50$) the difference became appalling, as shown in Fig. 3a: pre-training via mixture-based objectives lead to faster learning and higher performances, while pre-training via disjoint objectives turned out to be even *harmful* (Fig. 3b). This was motivated by the fact that the disjoint objective overfitted the task over the states reachable without coordinated exploration, resulting in almost deterministic policies, as shown in Fig. 5 in Appendix C. Finally, we tested the zero-shot capabilities of policy pre-training on the simpler but high dimensional exploration task of Env. (**ii**), where the goal was sampled randomly between *worst-case positions* at the boundaries of the region reachable by the arm. As shown in Fig. 4p, both joint and mixture were able to guarantee zero-shot performances via pre-training compatible with MA-TRPO after learning over 2e4 samples, while disjoint objectives were not. On the other hand, pre-training with joint objectives showed an extremely high-variance, leading to worst-case performances not better than the ones of random initialization. Mixture objectives on the other hand showed higher stability in guaranteeing compelling zero-shot performance. These results are the first to extend findings from single-agent environments [Zisselman et al., 2023] to multi-agent ones.

---

[13]While it is true that mixture objectives optimization appears to lead to slower optimization, this is the result of such pathological behaviors.

**Takeaways.** Overall, the proposed experiments managed to answer to all of the experimental questions: (**Q4.1**) Algorithm 1 is indeed able to optimize for finite-trial objectives; (**Q4.2**) **Mixture objectives enforce coordination**, essential when high efficiency is required, while joint or disjoint objectives may fail to lead to relevant solutions because of under or over optimization; (**Q4.3**) **The efficient coordination** through mixture objectives enforces the ability of **pre-training via state entropy maximization** to lead to **faster and better training** and even **zero-shot generalization**.

## 7 Related Works

Below, we summarize the most relevant work investigating related concepts.

**Entropic Functionals in MARL.** A large plethora of works on both swarm robotics [McLurkin and Yamins, 2005, Breitenmoser et al., 2010] and multi-agent intrinsic motivation, such as [Iqbal and Sha, 2019, Yang et al., 2021, Zhang et al., 2021b, 2023, Xu et al., 2024, Toquebiau et al., 2024], investigated the effects of employing entropic-like functions to boost exploration and performances in down-stream tasks. Importantly, these works are of empirical nature, and they do not investigate the theoretical properties of cMGs or multi-agent state entropy maximization, nor they propose algorithms able to pre-train policies *without access* to extrinsic rewards.[14] Finally, while a similar notion of cMGs was proposed in [Gemp et al., 2024, Kalogiannis et al., 2025], their contributions are focused on the existence and computation of equilibria and performance of centralized algorithms over infinite-trials objectives.

**State Entropy Maximization.** Entropy maximization in MDPs was first introduced in Hazan et al. [2019] and then investigated extensively in various subsequent works [e.g., Mutti and Restelli, 2020, Mutti et al., 2021, 2022b,c, Mutti, 2023, Liu and Abbeel, 2021b,a, Seo et al., 2021, Yarats et al., 2021, Zhang et al., 2021a, Guo et al., 2021, Yuan et al., 2022, Nedergaard and Cook, 2022, Yang and Spaan, 2023, Tiapkin et al., 2023, Jain et al., 2023, Kim et al., 2023, Zisselman et al., 2023, Li et al., 2024, Bolland et al., 2024, Zamboni et al., 2024b,a, De Paola et al., 2025]. Its infinite-trials formulation[15] can also be seen as a particular reward-free instance of state-entropy regularized MDPs [Brekelmans et al., 2022, Ashlag et al., 2025], although this reduction does not alleviate the aforementioned criticalities in solving such problems in multi-agent settings. To the best of our knowledge, our work is the first to study a multi-agent variation of the state entropy maximization problem.

**Policy Optimization.** Finally, our algorithmic solution (Algorithm 1) draws heavily on the literature of policy optimization and trust-region methods [Schulman et al., 2015]. Specifically, we considered an IS policy gradient estimator, which is partially inspired by the work of Metelli et al. [2020], but considers other forms of IS estimators, such as non-parametric k-NN estimators previously employed in Mutti et al. [2021].

## 8 Conclusions and Perspectives

In this paper, we introduce a principled framework for unsupervised pre-training in MARL via state entropy maximization. First, we formalize the problem as a convex generalization of Markov Games, and show that it can be defined via several different objectives: one can look at the joint distribution among all the agents, the marginals which are agent-specific, or the mixture which is a tradeoff of the two. Thus, we link these three options via performance bounds and we theoretically characterize how the problem, even when tractable in theory, is non-trivial in practice. Then, we design a practical algorithm and we use it in a set of experiments to confirm the expected superiority of mixture objectives in practice, due to their ability to enforce efficient coordination over short horizons. Future works can build over our results in many directions, for instance by pushing forward the knowledge on convex Markov Games, developing scalable algorithms for continuous domains, or performing extensive empirical investigation over large scale problems. We believe that our work can be a crucial step in the direction of extending policy pre-training via state entropy maximization in a principled way to yet more practical settings.

---

[14]The interested reader can refer to Mutti et al. [2021], Liu and Abbeel [2021b] for an extensive investigation of the fundamental differences between intrinsic motivation and state entropy maximization.

[15]Conversely, the finite-trial formulation targeted by Algorithm 1 is not studied in the literature of regularized MDPs.

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

# A  Further Insights on the Unsupervised Objectives.

**Motivating Example.**    Let us envision a team of agents in a "search and rescue" task. In a specific building (environment) the target may be found in different place (different rewards) and the unsupervised pre-training phase aims to prepare for all of them. Mixture entropy is a good surrogate objective in this case, as the agents will split up into different portions of the buildings to traverse in order to find the target quickly.

**Clarification on the Ideal Objective: Joint and Mixture Objectives Comparison**

As in single-agent settings, the goal of unsupervised (MA)RL via state entropy pre-training is to learn exploration for any possible task while interacting with a reward-free environment. If the tasks is assumed to be represented through state-based reward functions, the latter translates into state coverage: The state entropy is a proxy for state coverage [Hazan et al., 2019, Mutti et al., 2021, Liu and Abbeel, 2021b].

As a consequence, the most natural state entropy formulation in Markov games is the **joint state entropy**. However, it comes with some important drawbacks:

- **Estimation.** The support of the entropy grows exponentially with the number of agents $|S|^{|\mathcal{N}|}$, so does the complexity of the entropy estimation problem [Beirlant et al., 1997];

- **Concentration.** The empirical entropy concentrates as $\sqrt{K^{-1}}$ for $K$ trajectories (see Thm. 4.2);

- **Redundancy.** When Asm. 3.1 holds and the state space $|\mathcal{S}|$ is the same for every agent, the joint entropy may inflate state coverage as $(s, s')$ and $(s', s)$ are different joint states.

In other words, the problem of optimizing the joint entropy suffers from the *curse of multiagency*, which is particularly relevant in practice (while their difference might not be so relevant in ideal settings, see Fact 4.1 and Thm. B.6).

Another potential formulation is the **mixture state entropy**, which has the following properties:

- **Estimation.** The support of the entropy and therefore the estimation complexity do not grow with the number of agents;

- **Concentration.** The empirical entropy concentrates as $\sqrt{(K|\mathcal{N}|)^{-1}}$ for $K$ trajectories (see Thm. 4.2);

- **Redundancy.** For the mixture entropy objective, the joint states $(s, s')$ and $(s', s)$ are contributing in the same way; therefore, there is no difference in visiting one or the other.

The latter can be a limitation when we aim to explore all the possible *joint states*, e.g., when the reward functions of the agents will be different in the eventual tasks. Yet, at least the mixture entropy is also a lower bound to the joint entropy objective with a $\log(|\mathcal{N}|)$ approximation (see Lem. 4.1) and thus a valid proxy also in the latter case, given the favorable estimation and concentration properties.

# B  Proofs of the Main Theoretical Results

In this Section, we report the full proofing steps of the Theorems and Lemmas in the main paper.

**Lemma 4.1** (Entropy Mismatch). *For every cMG $\mathcal{M}_H$, for a fixed (joint) policy $\pi = (\pi^i)_{i \in \mathcal{N}}$ the infinite-trials objectives are ordered according to:*

$$\frac{H(d^\pi)}{|\mathcal{N}|} \leqslant \frac{1}{|\mathcal{N}|} \sum_{i \in [\mathcal{N}]} H(d_i^\pi) \leqslant H(\tilde{d}^\pi) \leqslant \sup_{i \in [\mathcal{N}]} H(d_i^\pi) + \log(|\mathcal{N}|) \leqslant H(d^\pi) + \log(|\mathcal{N}|)$$

*Proof.* The bounds follow directly from simple yet fundamental relationships between entropies of joint, marginal and mixture distributions which can be found in Paninski [2003], Kolchinsky and

Tracey [2017], in particular:

$$\frac{1}{|\mathcal{N}|}H(d^\pi) \leqslant \frac{1}{|\mathcal{N}|}\sum_{i\in[\mathcal{N}]}H(d_i^\pi) \overset{(a)}{\leqslant} H(\tilde{d}^\pi) \overset{(b)}{\leqslant} \frac{1}{|\mathcal{N}|}\sum_{i\in[\mathcal{N}]}H(d_i^\pi) + \log(|\mathcal{N}|)$$

$$\overset{(c)}{\leqslant} \sup_{i\in[\mathcal{N}]} H(d_i^\pi) + \log(|\mathcal{N}|) \leqslant H(d^\pi) + \log(|\mathcal{N}|)$$

where step (a) and (b) use the fact that $\tilde{d}^\pi(s) := \frac{1}{|\mathcal{N}|}\sum_{i\in[\mathcal{N}]}d_i^\pi(s)$ is a uniform mixture over the agents, whose distribution over the weights has entropy $\log(|\mathcal{N}|)$, so as we can apply the bounds from Kolchinsky and Tracey [2017]. Step (c) uses the fact that $H(d^\pi) = \sum_{i\in[\mathcal{N}]}H(d_i^\pi|d_{<i}^\pi)$, then taking the supremum as first $i$ it follows that $\sup_{i\in[\mathcal{N}]}H(d_i^\pi) = H(d^\pi) - \sum_{j\in[\mathcal{N}]>i}H(d_j^\pi|d_{<j}^\pi, d_i^\pi) \leqslant H(d^\pi)$ due to non-negativity of entropy. $\qquad\square$

**Theorem 4.2** (Finite-Trials Mismatch in cMGs). *For every cMG $\mathcal{M}_\mathcal{F}$ equipped with a L-Lipschitz function $\mathcal{F}$, let $K \in \mathbb{N}^+$ be a number of evaluation episodes/trials, and let $\delta \in (0, 1]$ be a confidence level, then for any (joint) policy $\pi = (\pi^i \in \Pi^i)_{i\in[\mathcal{N}]}$, it holds that*

$$|\zeta_K(\pi) - \zeta_\infty(\pi)| \leqslant LT\sqrt{\frac{2|\mathcal{S}|\log(2T/\delta)}{K}}, \quad |\zeta_K^i(\pi) - \zeta_\infty^i(\pi)| \leqslant LT\sqrt{\frac{2|\tilde{\mathcal{S}}|\log(2T/\delta)}{K}},$$

$$|\tilde{\zeta}_K(\pi) - \tilde{\zeta}_\infty(\pi)| \leqslant LT\sqrt{\frac{2|\tilde{\mathcal{S}}|\log(2T/\delta)}{|\mathcal{N}|K}}.$$

*Proof.* For the general proof structure, we adapt the steps of Mutti et al. [2022a] for cMDPs to the different objectives possible in cMGs. Let us start by considering joint objectives, then:

$$\left|\zeta_K(\pi) - \zeta_\infty(\pi)\right| = \left|\mathop{\mathbb{E}}_{d_K\sim p_K^\pi}[\mathcal{F}(d_K)] - \mathcal{F}(d^\pi)\right| \leqslant \mathop{\mathbb{E}}_{d_K\sim p_K^\pi}[|\mathcal{F}(d_K) - \mathcal{F}(d^\pi)|]$$

$$\overset{(a)}{\leqslant} \mathop{\mathbb{E}}_{d_K\sim p_K^\pi}[L\|d_K - d^\pi\|_1] \leqslant L\mathop{\mathbb{E}}_{d_K\sim p_K^\pi}[\|d_K - d^\pi\|_1]$$

$$\overset{(b)}{\leqslant} L\mathop{\mathbb{E}}_{d_K\sim p_K^\pi}\left[\max_{t\in[T]}\|d_{K,t} - d_t^\pi\|_1\right],$$

where in step (a) we apply the Lipschitz assumption on $\mathcal{F}$ to write and in step (b) we apply a maximization over the episode's step by noting that $d_K = \frac{1}{T}\sum_{t\in[T]}d_{K,t}$ and $d^\pi = \frac{1}{T}\sum_{t\in[T]}d_t^\pi$. We then apply bounds in high probability

$$Pr\left(\max_{t\in[T]}\|d_{K,t} - d_t^\pi\|_1 \geqslant \epsilon\right) \leqslant Pr\left(\bigcup_t \|d_{K,t} - d_t^\pi\|_1 \geqslant \epsilon\right)$$

$$\overset{(c)}{\leqslant} \sum_t Pr\left(\|d_{K,t} - d_t^\pi\|_1 \geqslant \epsilon\right)$$

$$\leqslant T\, Pr\left(\|d_{K,t} - d_t^\pi\|_1 \geqslant \epsilon\right),$$

with $\epsilon > 0$ and in step (c) we applied a union bound. We then consider standard concentration inequalities for empirical distributions [Weissman et al., 2003] so to obtain the final bound

$$Pr\left(\|d_{K,t} - d_t^\pi\|_1 \geqslant \sqrt{\frac{2|\mathcal{S}|\log(2/\delta')}{K}}\right) \leqslant \delta'. \tag{5}$$

By setting $\delta' = \delta/T$, and then plugging the empirical concentration inequality, we have that with probability at least $1 - \delta$

$$\left|\zeta_K(\pi) - \zeta_\infty(\pi)\right| \leqslant LT\sqrt{\frac{2|\mathcal{S}|\log(2T/\delta)}{K}},$$

which concludes the proof for joint objectives.

The proof for disjoint objectives follows the same rational by bounding each per-agent term separately and after noticing that due to Assumption 3.1, the resulting bounds get simplified in the overall averaging. As for mixture objectives, the only core difference is after step (b), where $\tilde{d}_K$ takes the place of $d_K$ and $\tilde{d}^\pi$ of $d^\pi$. The remaining steps follow the same logic, out of noticing that the empirical distribution with respect to $\tilde{d}^\pi$ is taken with respect $|\mathcal{N}|K$ samples in total. Both the two bounds then take into account that the support of the empirical distributions have size $|\tilde{\mathcal{S}}|$ and not $|\mathcal{S}|$. $\qquad\square$

## B.1 Policy Gradient in cMGs with Infinite-Trials Formulations.

In this Section, we analyze policy search for the infinite-trials joint problem $\zeta_\infty$ of Eq. (1), via projected gradient ascent over parametrized policies, providing in Th. B.6 the formal counterpart of Fact 4.1 in the Main paper. As a side note, all of the following results hold for the (infinite-trials) mixture objective $\tilde{\zeta}_\infty$ of Eq. (3). We will consider the class of parametrized policies with parameters $\theta_i \in \Theta_i \subset \mathbb{R}^d$, with the joint policy then defined as $\pi_\theta, \theta \in \Theta = \times_{i\in[\mathcal{N}]}\Theta_i$. Additionally, we will focus on the computational complexity only, by assuming access to the exact gradient. The study of statistical complexity surpasses the scope of the current work. We define the **(independent) Policy Gradient Ascent** (PGA) update as:

$$\theta_i^{k+1} = \arg\max_{\theta_i\in\Theta_i} \zeta_\infty(\pi_{\theta^k}) + \langle\nabla_{\theta_i}\zeta_\infty(\pi_{\theta^k}), \theta_i - \theta_i^k\rangle - \frac{1}{2\eta}\|\theta_i - \theta_i^k\|^2 = \Pi_{\Theta_i}\{\theta_i^k + \eta\nabla_{\theta_i}\zeta_\infty(\pi_{\theta^k})\} \quad (6)$$

where $\Pi_{\Theta_i}\{\cdot\}$ denotes Euclidean projection onto $\Theta_i$, and equivalence holds by the convexity of $\Theta_i$. The classes of policies that allow for this condition to be true will be discussed shortly.

In general the overall proof is built of three main steps, shared with the theory of Potential Markov Games [Leonardos et al., 2022]: (i) prove the existence of well behaved stationary points; (ii) prove that performing independent policy gradient is equivalent to perform joint policy gradient; (iii) prove that the (joint) PGA update converges to the stationary points via single-agent like analysis. In order to derive the subsequent convergence proof, we will make the following assumptions:

**Assumption B.1.** Define the quantity $\lambda(\theta) := d^{\pi_\theta}$, then:
**(i).** $\lambda(\cdot)$ forms a bijection between $\Theta$ and $\lambda(\Theta)$, where $\Theta$ and $\lambda(\Theta)$ are closed and convex.
**(ii).** The Jacobian matrix $\nabla_\theta\lambda(\theta)$ is Lipschitz continuous in $\Theta$.
**(iii).** Denote $g(\cdot) := \lambda^{-1}(\cdot)$ as the inverse mapping of $\lambda(\cdot)$. Then there exists $\ell_\theta > 0$ s.t. $\|g(\lambda) - g(\lambda')\| \leqslant \ell_\theta\|\lambda - \lambda'\|$ for some norm $\|\cdot\|$ and for all $\lambda, \lambda' \in \lambda(\Theta)$.

**Assumption B.2.** There exists $L > 0$ such that the gradient $\nabla_\theta\zeta_\infty(\pi_\theta)$ is $L$-Lipschitz.

**Assumption B.3.** The agents have access to a gradient oracle $\mathcal{O}(\cdot)$ that returns $\nabla_{\theta_i}\zeta_\infty(\pi_\theta)$ for any deployed joint policy $\pi_\theta$.

**On the Validity of Assumption B.1.** This set of assumptions enforces the objective $\zeta_\infty(\pi_\theta)$ to be well-behaved with respect to $\theta$ even if non-convex in general, and will allow for a rather strong result. Yet, the assumptions are known to be true for directly parametrized policies over the whole support of the distribution $d^\pi$ [Zhang et al., 2020], and as a result they implicitly require agents to employ policies conditioned over the full state-space $\mathcal{S}$. Fortunately enough, they also guarantee $\Theta$ to be convex.

**Lemma B.4 ((i)** Global optimality of stationary policies [Zhang et al., 2020]**).** *Suppose Assumption B.1 holds, and $\mathcal{F}$ is a concave, and continuous function defined in an open neighborhood containing $\lambda(\Theta)$. Let $\theta^*$ be a first-order stationary point of problem (1), i.e.,*

$$\exists u^* \in \hat{\partial}(\mathcal{F}\circ\lambda)(\theta^*), \quad s.t. \quad \langle u^*, \theta - \theta^*\rangle \leqslant 0 \quad for \quad \forall\theta\in\Theta. \quad (7)$$

*Then $\theta^*$ is a globally optimal solution of problem (1).*

This result characterizes the optimality of stationary points for Eq. (1). Furthermore, we know from Leonardos et al. [2022] that stationary points of the objective are Nash Equilibria.

**Lemma B.5 ((ii)** Projection Operator [Leonardos et al., 2022]**).** *Let $\theta := (\theta_1, ..., \theta_\mathcal{N})$ be the parameter profile for all agents and use the update of Eq. (6) over a non-disjoint infinite-trials objective. Then, it holds that*

$$\Pi_\Theta\{\theta^k + \eta\nabla_\theta\zeta_\infty(\pi_{\theta^k})\} = \left(\Pi_{\Theta_i}\{\theta_i^k + \eta\nabla_{\theta_i}\zeta_\infty(\pi_{\theta^k})\}\right)_{i\in[\mathcal{N}]}$$

This result will only be used for the sake of the convergence analysis, since it allows to analyze independent updates as joint updates over a single objective. The following Theorem is the formal counterpart of Fact 4.1 and it is a direct adaptation to the multi-agent case of the single-agent proof by Zhang et al. [2020], by exploiting the previous result.

**Theorem B.6 ((iii)** Convergence rate of independent PGA to stationary points (Formal Fact 4.1)**).** *Let Assumptions B.1 and B.2 hold. Denote $D_\lambda := \max_{\lambda,\lambda' \in \lambda(\Theta)} \|\lambda - \lambda'\|$ as defined in Assumption B.1(iii). Then the independent policy gradient update (6) with $\eta = 1/L$ satisfies for all $k$ with respect to a stationary (joint) policy $\pi_{\theta*}$ the following*

$$\zeta_\infty(\pi_{\theta*}) - \zeta_\infty(\pi_{\theta^k}) \leqslant \frac{4L\ell_\theta^2 D_\lambda^2}{k+1}.$$

*Proof.* First, the Lipschitz continuity in Assumption B.2 indicates that

$$\left| \zeta_\infty(\lambda(\theta)) - \zeta_\infty(\lambda(\theta^k)) - \langle \nabla_\theta \zeta_\infty(\lambda(\theta^k)), \theta - \theta^k \rangle \right| \leqslant \frac{L}{2} \|\theta - \theta^k\|^2.$$

Consequently, for any $\theta \in \Theta$ we have the ascent property:

$$\zeta_\infty(\lambda(\theta)) \geqslant \zeta_\infty(\lambda(\theta^k)) + \langle \nabla_\theta \zeta_\infty(\lambda(\theta^k)), \theta - \theta^k \rangle - \frac{L}{2} \|\theta - \theta^k\|^2 \geqslant \zeta_\infty(\lambda(\theta)) - L\|\theta - \theta^k\|^2. \quad (8)$$

The optimality condition in the policy update rule (6) coupled with the result of Lemma B.5 allows us to follow the same rational as Zhang et al. [2020]. We will report their proof structure after this step for completeness.

$$\zeta_\infty(\lambda(\theta^{k+1})) \geqslant \zeta_\infty(\lambda(\theta^k)) + \langle \nabla_\theta \zeta_\infty(\lambda(\theta^k)), \theta^{k+1} - \theta^k \rangle - \frac{L}{2} \|\theta^{k+1} - \theta^k\|^2$$

$$= \max_{\theta \in \Theta} \zeta_\infty(\lambda(\theta^k)) + \langle \nabla_\theta \zeta_\infty(\lambda(\theta^k)), \theta - \theta^k \rangle - \frac{L}{2} \|\theta - \theta^k\|^2$$

$$\overset{(a)}{\geqslant} \max_{\theta \in \Theta} \zeta_\infty(\lambda(\theta)) - L\|\theta - \theta^k\|^2$$

$$\overset{(b)}{\geqslant} \max_{\alpha \in [0,1]} \left\{ \zeta_\infty(\lambda(\theta_\alpha)) - L\|\theta_\alpha - \theta^k\|^2 : \theta_\alpha = g(\alpha\lambda(\theta^*) + (1-\alpha)\lambda(\theta^k)) \right\}. \quad (9)$$

where step (a) follows from (8) and step (b) uses the convexity of $\lambda(\Theta)$. Then, by the concavity of $\zeta_\infty$ and the fact that the composition $\lambda \circ g = id$ due to Assumption B.1(i), we have that:

$$\zeta_\infty(\lambda(\theta_\alpha)) = \zeta_\infty(\alpha\lambda(\theta^*) + (1-\alpha)\lambda(\theta^k)) \geqslant \alpha\zeta_\infty(\lambda(\theta^*)) + (1-\alpha)\zeta_\infty(\lambda(\theta^k)).$$

Moreover, due to Assumption B.1(iii) we have that:

$$\begin{aligned} \|\theta_\alpha - \theta^k\|^2 &= \|g(\alpha\lambda(\theta^*) + (1-\alpha)\lambda(\theta^k)) - g(\lambda(\theta^k))\|^2 \quad (10) \\ &\leqslant \alpha^2\ell_\theta^2 \|\lambda(\theta^*) - \lambda(\theta^k)\|^2 \\ &\leqslant \alpha^2\ell_\theta^2 D_\lambda^2. \end{aligned}$$

From which we get

$$\zeta_\infty(\lambda(\theta^*)) - \zeta_\infty(\lambda(\theta^{k+1}))$$

$$\leqslant \min_{\alpha \in [0,1]} \left\{ \zeta_\infty(\lambda(\theta^*)) - \zeta_\infty(\lambda(\theta_\alpha)) + L\|\theta_\alpha - \theta^k\|^2 : \theta_\alpha = g(\alpha\lambda(\theta^*) + (1-\alpha)\lambda(\theta^k)) \right\}$$

$$\leqslant \min_{\alpha \in [0,1]} (1-\alpha)\big(\zeta_\infty(\lambda(\theta^*)) - \zeta_\infty(\lambda(\theta^k))\big) + \alpha^2 L\ell_\theta^2 D_\lambda^2. \quad (11)$$

We define $\Lambda(\pi_\theta) := \lambda(\theta)$, then $\alpha_k = \frac{\zeta_\infty(\Lambda(\pi^*)) - \zeta_\infty(\Lambda(\pi^k))}{2L\ell_\theta^2 D_\lambda^2} \geqslant 0$, which is the minimizer of the RHS of (11) as long as it satisfies $\alpha_k \leqslant 1$. Now, we claim the following: If $\alpha_k \geqslant 1$ then $\alpha_{k+1} < 1$. Further, if $\alpha_k < 1$ then $\alpha_{k+1} \leqslant \alpha_k$. The two claims together mean that $(\alpha_k)_k$ is decreasing and all $\alpha_k$ are in $[0,1)$ except perhaps $\alpha_0$.

To prove the first of the two claims, assume $\alpha_k \geqslant 1$. This implies that $\zeta_\infty(\Lambda(\pi^*)) - \zeta_\infty(\Lambda(\pi^k)) \geqslant 2L\ell_\theta^2 D_\lambda^2$. Hence, choosing $\alpha = 1$ in (11), we get

$$\zeta_\infty(\lambda(\theta^*)) - \zeta_\infty(\lambda(\theta^k)) \leqslant L\ell_\theta^2 D_\lambda^2$$

which implies that $\alpha_{k+1} \leqslant 1/2 < 1$. To prove the second claim, we plug $\alpha_k$ into (11) to get

$$\zeta_\infty(\lambda(\theta^*)) - \zeta_\infty(\lambda(\theta^{k+1})) \leqslant \left(1 - \frac{\zeta_\infty(\lambda(\theta^*)) - \zeta_\infty(\lambda(\theta^k))}{4L\ell_\theta^2 D_\lambda^2}\right)(\zeta_\infty(\lambda(\theta^*)) - \zeta_\infty(\lambda(\theta^k))),$$

which shows that $\alpha_{k+1} \leqslant \alpha_k$ as required.

Now, by our preceding discussion, for $k = 1, 2, \dots$ the previous recursion holds. Using the definition of $\alpha_k$, we rewrite this in the equivalent form

$$\frac{\alpha_{k+1}}{2} \leqslant \left(1 - \frac{\alpha_k}{2}\right) \cdot \frac{\alpha_k}{2}.$$

By rearranging the preceding expressions and algebraic manipulations, we obtain

$$\frac{2}{\alpha_{k+1}} \geqslant \frac{1}{\left(1 - \frac{\alpha_k}{2}\right) \cdot \frac{\alpha_k}{2}} = \frac{2}{\alpha_k} + \frac{1}{1 - \frac{\alpha_k}{2}} \geqslant \frac{2}{\alpha_k} + 1.$$

For simplicity assume that $\alpha_0 < 1$ also holds. Then, $\frac{2}{\alpha_k} \geqslant \frac{2}{\alpha_0} + k$, and consequenlty

$$\zeta_\infty(\lambda(\theta^*)) - \zeta_\infty(\lambda(\theta^k)) \leqslant \frac{\zeta_\infty(\lambda(\theta^*)) - \zeta_\infty(\lambda(\theta^0))}{1 + \frac{\zeta_\infty(\lambda(\theta^*)) - \zeta_\infty(\lambda(\theta^0))}{4L\ell_\theta^2 D_\lambda^2} \cdot k} \leqslant \frac{4L\ell_\theta^2 D_\lambda^2}{k}.$$

A similar analysis holds when $\alpha_0 > 1$. Combining these two gives that $\zeta_\infty(\lambda(\pi^*)) - \zeta_\infty(\lambda(\pi^k)) \leqslant \frac{4L\ell_\theta^2 D_\lambda^2}{k+1}$ no matter the value of $\alpha_0$, which proves the result. $\qquad \square$

## B.2 The Use of Markovian and Non-Markovian Policies in cMGs with Finite-Trials Formulations.

The following result describes how in cMGs, as for cMDPs, Non-Markovian policies are the right policy class to employ to guarantee well-behaved results.

**Lemma B.1** (Sufficiency of Disjoint Non-Markvoian Policies). *For every cMG $\mathcal{M}$ there exist a joint policy $\pi^\star = (\pi^{\star,i})_{i \in \mathcal{N}}$, with $\pi^{\star,i} \in \Delta_{\mathcal{S}^T}^{\mathcal{A}^i}$ being a deterministic Non-Markovian policy, that is a Nash Equilibrium for non-Disjoint single-trial objectives, for $K = 1$.*

*Proof.* The proof builds over a straight reduction. We build from the original MG $\mathcal{M}$ a temporally extended Markov Game $\tilde{\mathcal{M}} = (\mathcal{N}, \tilde{\mathcal{S}}, \mathcal{A}, \mathbb{P}, r, \mu, T)$. A state $\tilde{s}$ is defined for each history that can be induced, i.e., $\tilde{s} \in \tilde{\mathcal{S}} \iff \mathbf{s} \in \mathcal{S}^T$. We keep the other objects equivalent, where for the extended transition model we solely consider the last state in the history to define the conditional probability to the next history. We introduce a common reward function across all the agents $r : \tilde{\mathcal{S}} \to \mathbb{R}$ such that $r(\tilde{s}) = H(d(\tilde{s}))$ for joint objectives and $r(\tilde{s}) = (1/N) \sum_{i \in [\mathcal{N}]} H(d_i(\tilde{s}_i))$ for mixture objectives, for all the histories of length T and 0 otherwise. We now know that according to Leonardos et al. [Theorem 3.1, 2022] there exists a deterministic Markovian policy $\tilde{\pi}^\star = (\tilde{\pi}^i)_{i \in \mathcal{N}}, \tilde{\pi}^i \in \Delta_{\tilde{\mathcal{S}}}^{\mathcal{A}_i}$ that is a Nash Equilibrium for $\tilde{\mathcal{M}}$. Since $\tilde{s}$ corresponds to the set of histories of the original game, $\tilde{\pi}^\star$ maps to a non-Markovian policy in it. Finally, it is straightforward to notice that the NE of $\tilde{\pi}^\star$ for $\tilde{\mathcal{M}}$ implies the NE of $\tilde{\pi}^\star$ for the original cMG $\mathcal{M}$. $\qquad \square$

The previous result implicitly asks for policies conditioned over the joint state space, as happened for infinite-trials objectives as well. Interestingly, finite-trials objectives allow for a further characterization of how an optimal Markovian policy would behave when conditioned on the per-agent states only:

**Lemma B.7** (Behavior of Optimal Markovian Decentralized Policies). *Let $\pi_{NM} = (\pi_{NM}^i \in \Delta_{\mathcal{S}^T}^{\mathcal{A}^i})_{i \in [\mathcal{N}]}$ an optimal deterministic non-Markovian centralized policy and $\bar{\pi}_M = (\bar{\pi}_M^i \in \Delta_{\mathcal{S}}^{\mathcal{A}^i})_{i \in [\mathcal{N}]}$ the optimal Markovian centralized policy, namely $\bar{\pi}_M = \arg\max_{\pi = (\pi^i \in \Delta_{\mathcal{S}}^{\mathcal{A}^i})_{i \in [\mathcal{N}]}} \zeta_1(\pi)$. For a fixed sequence $\mathbf{s}_t \in \mathcal{S}^t$ ending in state $s = (s_i, s_{-i})$, the variance of the event of the optimal Markovian decentralized policy $\pi_M = (\pi_M^i \in \Delta_{\mathcal{S}_i}^{\mathcal{A}^i})_{i \in [\mathcal{N}]}$ taking $a^* = \pi_{NM}(\cdot|\mathbf{s}_t) = \bar{\pi}_M(\cdot|s, t)$ in $s_i$ at step $t$ is given by*

$$\mathbb{Var}\left[\mathcal{B}(\pi_M(a^*|s_i, t))\right] = \mathbb{Var}_{\mathbf{s} \oplus s \sim p_t^{\pi_{NM}}}\left[\mathbb{E}\left[\mathcal{B}(\pi_{NM}(a^*|\mathbf{s} \oplus s))\right]\right]$$

$$+ \mathbb{Var}_{\mathbf{s} \oplus (\cdot, s_{-i}) \sim p_t^{\bar{\pi}_M}}\left[\mathbb{E}\left[\mathcal{B}(\bar{\pi}_M(a^*|s_i, s_{-i}, t))\right]\right].$$

*where* $\mathbf{s} \oplus s \in \mathcal{S}^t$ *is any sequence of length* $t$ *such that the final state is* $s$, *i.e.,* $\mathbf{s} \oplus s := (\mathbf{s}_{t-1} \in \mathcal{S}^{t-1}) \oplus s$, *and* $\mathcal{B}(x)$ *is a Bernoulli with parameter* $x$.

Unsurprisingly, this Lemma shows that whenever the optimal Non-Markovian strategy for requires to adapt its decision in a joint state $s$ according to the history that led to it, an optimal Markovian policy for the same objective must necessarily be a stochastic policy, additionally, whenever the optimal Markovian policy conditioned over per-agent states only will need to be stochastic whenever the optimal Markovian strategy conditioned on the full states randomizes its decision based on the joint state $s$.

*Proof.* Let us consider the random variable $A_i \sim \mathcal{P}_i$ denoting the event "the agent $i$ takes action $a_i^* \in \mathcal{A}_i$". Through the law of total variance Bertsekas and Tsitsiklis [2002], we can write the variance of $A$ given $s \in \mathcal{S}$ and $t \geq 0$ as

$$\mathbb{Var}\big[A|s,t\big] = \mathbb{E}\big[A^2|s,t\big] - \mathbb{E}\big[A|s,t\big]^2 = \mathbb{E}_{\mathbf{s}}\Big[\mathbb{E}\big[A^2|s,t,\mathbf{s}\big]\Big] - \mathbb{E}_{\mathbf{s}}\Big[\mathbb{E}\big[A|s,t,\mathbf{s}\big]\Big]^2$$

$$= \mathbb{E}_{\mathbf{s}}\Big[\mathbb{Var}\big[A|s,t,\mathbf{s}\big] + \mathbb{E}\big[A|s,t,\mathbf{s}\big]^2\Big] - \mathbb{E}_{\mathbf{s}}\Big[\mathbb{E}_{\pi}\big[A|s,t,\mathbf{s}\big]\Big]^2$$

$$= \mathbb{E}_{\mathbf{s}}\Big[\mathbb{Var}\big[A|s,t,\mathbf{s}\big]\Big] + \mathbb{E}_{\mathbf{s}}\Big[\mathbb{E}\big[A|s,t,\mathbf{s}\big]^2\Big] - \mathbb{E}_{\mathbf{s}}\Big[\mathbb{E}\big[A|s,t,\mathbf{s}\big]\Big]^2$$

$$= \mathbb{E}_{\mathbf{s}}\Big[\mathbb{Var}\big[A|s,t,\mathbf{s}\big]\Big] + \mathbb{Var}_{\mathbf{s}}\Big[\mathbb{E}\big[A|s,t,\mathbf{s}\big]\Big]. \tag{12}$$

Now let the conditioning event $\mathbf{s}$ be distributed as $\mathbf{s} \sim p_{t-1}^{\pi_{\mathrm{NM}}}$, so that the condition $s,t,\mathbf{s}$ becomes $\mathbf{s} \oplus s$ where $\mathbf{s} \oplus s = (s_0, a_0, s_1, \ldots, s_t = s) \in \mathcal{S}^t$, and let the variable $A$ be distributed according to $\mathcal{P}$ that maximizes the objective given the conditioning. Hence, we have that the variable $A$ on the left hand side of (12) is distributed as a Bernoulli $\mathcal{B}(\bar{\pi}_{\mathrm{M}}(a^*|s,t))$, and the variable $A$ on the right hand side of (13) is distributed as a Bernoulli $\mathcal{B}(\pi_{\mathrm{NM}}(a^*|\mathbf{s} \oplus s))$. Thus, we obtain

$$\mathbb{Var}\big[\mathcal{B}(\bar{\pi}_{\mathrm{M}}(a^*|s,t))\big] = \mathop{\mathbb{E}}_{\mathbf{s} \oplus s \sim p_t^{\pi_{\mathrm{NM}}}}\big[\mathbb{Var}\big[\mathcal{B}(\pi_{\mathrm{NM}}(a^*|\mathbf{s}\oplus s))\big]\big] + \mathop{\mathbb{Var}}_{\mathbf{s} \oplus s \sim p_t^{\pi_{\mathrm{NM}}}}\big[\mathbb{E}\big[\mathcal{B}(\pi_{\mathrm{NM}}(a^*|\mathbf{s}\oplus s))\big]\big].$$

$$\tag{13}$$

We know from Lemma B.1 that the policy $\pi_{\mathrm{NM}}$ is deterministic, so that $\mathbb{Var}\big[\mathcal{B}(\pi_{\mathrm{NM}}(a^*|\mathbf{s} \oplus s))\big] = 0$ for every $\mathbf{s} \oplus s$. We then repeat the same steps in order to compare the two different Markovian policies:

$$\mathbb{Var}\big[A|s_i,t\big] = \mathop{\mathbb{E}}_{s_{-i}}\Big[\mathbb{Var}\big[A|s_i,s_{-i},t\big]\Big] + \mathop{\mathbb{Var}}_{s_{-i}}\Big[\mathbb{E}\big[A|s_i,s_{-i},t\big]\Big].$$

Repeating the same considerations as before we get that we can use (13) to get:

$$\mathbb{Var}\big[\mathcal{B}(\pi_{\mathrm{M}}(a^*|s_i,t))\big] = \mathop{\mathbb{E}}_{\mathbf{s} \oplus (\cdot, s_{-i}) \sim p_t^{\bar{\pi}_{\mathrm{M}}}}\Big[\mathbb{Var}\big[\mathcal{B}(\bar{\pi}_{\mathrm{M}}(a^*|s_i,s_{-i},t))\big] + \mathbb{E}\big[\mathcal{B}(\bar{\pi}_{\mathrm{M}}(a^*|s_i,s_{-i},t))\big]\Big]$$

$$= \mathop{\mathbb{Var}}_{\mathbf{s} \oplus s \sim p_t^{\pi_{\mathrm{NM}}}}\big[\mathbb{E}\big[\mathcal{B}(\pi_{\mathrm{NM}}(a^*|\mathbf{s} \oplus s))\big]\big] + \mathop{\mathbb{Var}}_{\mathbf{s} \oplus (\cdot, s_{-i}) \sim p_t^{\bar{\pi}_{\mathrm{M}}}}\big[\mathbb{E}\big[\mathcal{B}(\bar{\pi}_{\mathrm{M}}(a^*|s_i,s_{-i},t))\big]\big].$$

$\square$

## C   Details on the Empirical Corroboration.

All the experiments were performed over an Apple M2 chip (8-core CPU, 8-core GPU, 16-core Neural Engine) with 8 GB unified memory with a maximum time of execution of 24 hours.

**Environments.**   The main empirical proof of concept was based on two environments. First, Env. (**i**), the so called *secret room* environment by Liu et al. [2021]. In this environment, two agents operate within two rooms of a $10 \times 10$ discrete grid. There is one switch in each room, one in position $(1,9)$ (corner of first room), another in position $(9,1)$ (corner of second room). The rooms are separated by a door and agents start in the same room deterministically at positions $(1,1)$ and $(2,2)$ respectively. The door will open only when one of the switches is occupied, which means that the (Manhattan) distance between one of the agents and the switch is less than $1.5$. The full state vector contains $x, y$ locations of the two agents and binary variables to indicate if doors are open *but* per-agent policies are

conditioned on their respective states only and the state of the door. For Sparse-Rewards Tasks, the goal was set to be deterministically at the worst case, namely $(9, 9)$ and to provide a positive reward to both the agents of $100$ when reached, which means again that the (Manhattan) distance between one of the agents and the switch is less than $1.5$, a reward of $0$ otherwise. The second environment, Env. (**ii**), was the MaMuJoCo *reacher* environment Peng et al. [2021]. In this environment, two agents operate the two linked joints and each space dimension is discretized over $10$ bins. Per-agent policies were conditioned on their respective joint angles only. For Sparse-Rewards Tasks, the goal was set to be randomly at the worst case, namely on position $(\pm 0.21, \pm 0.21)$ on the boundary of the reachable area. Reaching the goal mean to have a tip position (not observable by the agents and not discretized) at a distance less that $0.05$ and provides a positive reward to both the agents of $1$ when reached, a reward of $0$ otherwise.

**Class of Policies.** In Env. (**i**), the policy was parametrized by a dense $(64, 64)$ Neural Network that takes as input the per-agent state features and outputs an action vector probabilities through a last soft-max layer. In Env. (**ii**), the policy was represented by a Gaussian distribution with diagonal covariance matrix. It takes as input the environment state features and outputs an action vector. The mean is state-dependent and is the downstream output of a a dense $(64, 64)$ Neural Network. The standard deviation is state-independent, represented by a separated trainable vector and initialized to $-0.5$. The weights are initialized via Xavier Initialization.

**Trust Region Pure Exploration (TRPE).** As outlined in the pseudocode of Algorithm 1, in each epoch a dataset of $N$ trajectories is gathered for a given exploration horizon $T$, leading to the reported number of samples. Throughout the experiment the number of epochs $e$ were set equal to $e = 10k$, the number of trajectories $N = 10$, the KL threshold $\delta = 6$, the maximum number of off-policy iterations set to $n_{\text{off,iter}} = 20$, the learning rate was set to $\eta = 10^{-5}$ and the number of seeds set equal to $4$ due to the inherent low stochasticity of the environment.

**Multi-Agent TRPO (MA-TRPO).** We follow the same notation in Duan et al. [2016]. Agents have independent critics $(64, 64)$ Dense networks and in each epoch a dataset of $N$ trajectories is gathered for a given exploration horizon $T$ for each agent, leading to the reported number of samples. Throughout the experiment the number of epochs $e$ were set equal to $e = 100$, the number of trajectories building the batch size $N = 20$, the KL threshold $\delta = 10^{-4}$, the maximum number of off-policy iterations set to $n_{\text{off,iter}} = 20$, the discount was set to $\gamma = 0.99$.

The Repository is made available at the following Repository.

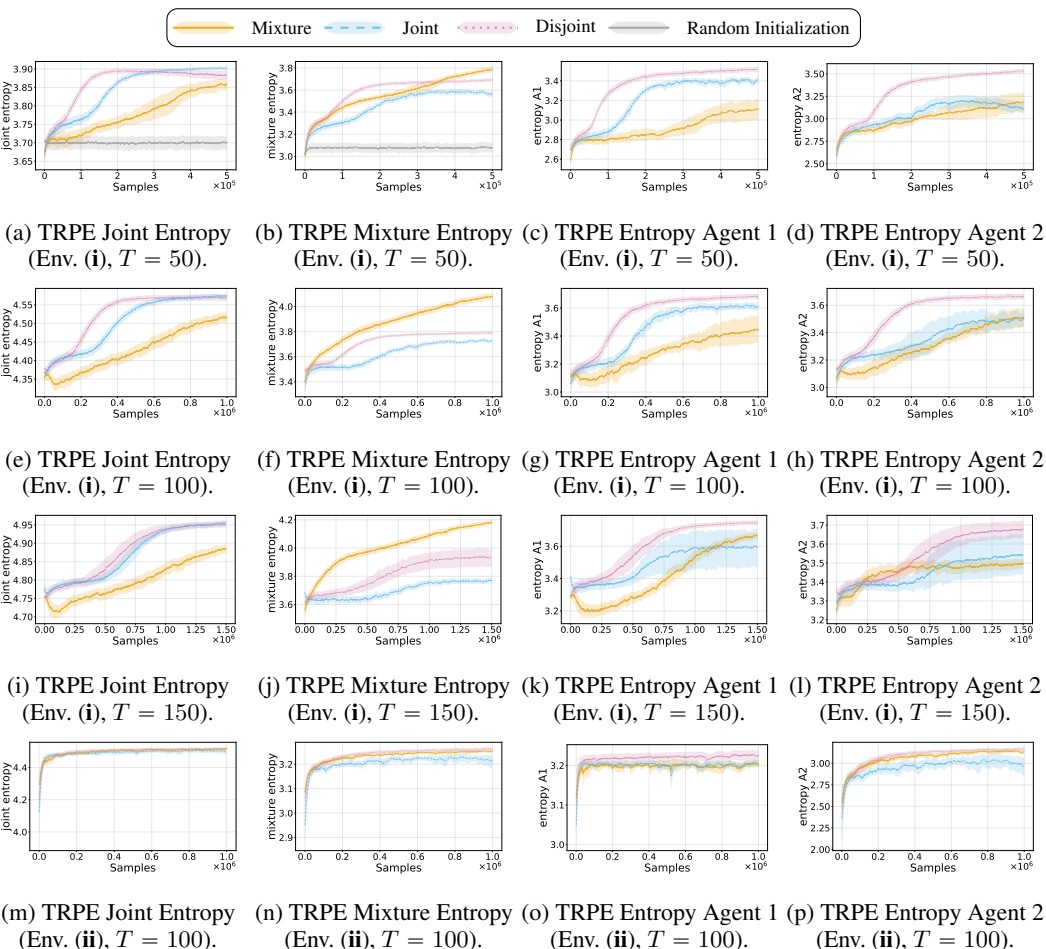

(a) TRPE Joint Entropy (Env. (**i**), $T = 50$).

(b) TRPE Mixture Entropy (Env. (**i**), $T = 50$).

(c) TRPE Entropy Agent 1 (Env. (**i**), $T = 50$).

(d) TRPE Entropy Agent 2 (Env. (**i**), $T = 50$).

(e) TRPE Joint Entropy (Env. (**i**), $T = 100$).

(f) TRPE Mixture Entropy (Env. (**i**), $T = 100$).

(g) TRPE Entropy Agent 1 (Env. (**i**), $T = 100$).

(h) TRPE Entropy Agent 2 (Env. (**i**), $T = 100$).

(i) TRPE Joint Entropy (Env. (**i**), $T = 150$).

(j) TRPE Mixture Entropy (Env. (**i**), $T = 150$).

(k) TRPE Entropy Agent 1 (Env. (**i**), $T = 150$).

(l) TRPE Entropy Agent 2 (Env. (**i**), $T = 150$).

(m) TRPE Joint Entropy (Env. (**ii**), $T = 100$).

(n) TRPE Mixture Entropy (Env. (**ii**), $T = 100$).

(o) TRPE Entropy Agent 1 (Env. (**ii**), $T = 100$).

(p) TRPE Entropy Agent 2 (Env. (**ii**), $T = 100$).

Figure 4: Full Visualization of Reported Experiments. Experiments with longer horizons highlight how **the easier the task, the less crucial the distinction between the objectives is.**

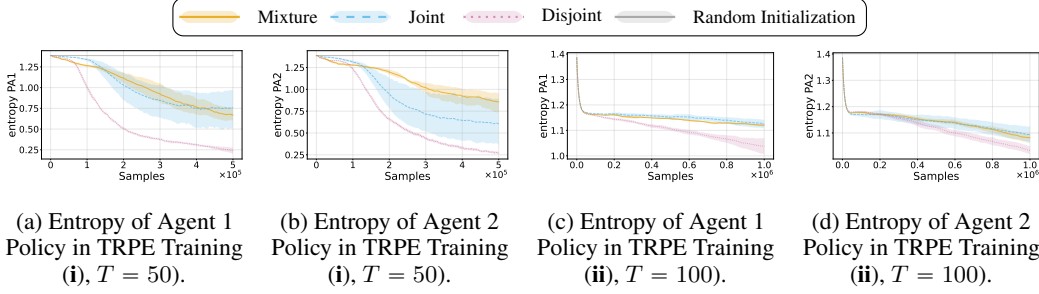

(a) Entropy of Agent 1 Policy in TRPE Training (**i**), $T = 50$.

(b) Entropy of Agent 2 Policy in TRPE Training (**i**), $T = 50$.

(c) Entropy of Agent 1 Policy in TRPE Training (**ii**), $T = 100$.

(d) Entropy of Agent 2 Policy in TRPE Training (**ii**), $T = 100$.

Figure 5: Policiy Entropy Insights for TRPO Pretraining in Env (**i**) and Env (**ii**). **Lower Entropic Policies with Disjoint Objectives might justify the difference in pre-training performance even if the performances in training are similar**.

