# OpenReview forum: "Towards Principled Unsupervised Multi-Agent Reinforcement Learning"
_NeurIPS.cc/2025/Conference — NeurIPS 2025 poster_

### Official Review · Reviewer_xd97 · 2025-06-18

**Clarity:** 3
**Significance:** 4
**Originality:** 3
**Rating:** 5
**Confidence:** 4

**Summary:**

The authors investigate unsupervised pre-training in multi-agent setting through the maximization of the entropy of the state distribution. They use convex Markov games to study three different settings: the joint objective (which simply looks at the entropy of the joint state), the disjoint objective (which considers maximizing each agent's entropy separately) and the mixture objective (which considers the entropy of a uniform mixture of the state distribution). The author argues the joint objective is intractible in practice while the disjoint doesn't optimize for coordination. Instead, the mixture objective is both tractable and encourages coordination. They show bounds relating these three objectives and particularly how the growing joint state space is problematic for the finite trial setting.

Then, the authors propose Trust Region Pure Exploration (TRPE) a decentralized algorithm for maximizing general convex functions in multi-agent settings with finite trials. The algorithm maximizes the parameters of each policy separately by maximizing a surrogate function constrained by a KL with the previous policy. Empirically, they show that TRPE with a mixture objective enables better exploration in a secret room setting. Finally, in sparse-reward settings, the mixture objective allows for pretraining that enables both better zero-shot performance and transfer learning.

**Questions:**

- What is the difference between the convex Markov game setting and the regularized MDP setting with no rewards?
- Was the mixture objective/decomposition previously applied (e.g. in Kolchinsky and Tracey) or is it a contribution of this work?
- For TRPE, it is unclear to me what is the purpose of the trust region update and the surrogate function.
- You mention "as the convex nature of the problem hinders the applicability of Bellman operators". Could a soft Bellman operator be used?
- It is unclear to me what Fig. 2. (right) is illustrating? What is the clustering behavior?
- Do you have any hypotheses for why the zero-shot learning seem to do better than post-learning with a random init?

If my questions are answered and the clarity of Sec. 5 is improved, I would be willing to increase my score from 4 -> 5.

**Ethical Concerns:**

["NO or VERY MINOR ethics concerns only"]

**Final Justification:**

The paper is well-motivated and proposes a novel approach with interesting theoretical and experimental results. My concerns regarding some details and the clarity of certain sections have been largely addressed. I have thus raised my score to "Accept".

**Limitations:**

Yes.

**Paper Formatting Concerns:**

No issues.

**Quality:**

3

**Strengths And Weaknesses:**

**Strengths**
- The paper idea is well-motivated and addresses an important problem of entropy-based pre-training for multi-agent settings. It is indeed unclear what the best way to compute entropy would be and the paper analyzes/compares 3 relevant possibilities.
- The authors provide relevant theoretical results for their analysis and describe their shortcomings.
- The mixture objective has a compelling interpretation as maximizing individual entropy + KL with other policies. The theoretical advantages are also clear given Thm. 4.2 where the bound relies on $|\tilde{S}|$ instead of the joint state.

**Weaknesses**
- Some parts of the paper were less clear, particularly TRPE and the motivations behind its design.
- The work could benefit from further discussion of the connection with regularized MDPs (see question).
- Related work lacks detail contrasting contributions.
- Some typos/writing issues, for example:
  - "which we let discrete and finite with size" -> which we assume to be discrete and finite in size
  - "mixture objectives enforce a clustering behavior (Fig. 2e) and result in a better efficient exploration". Should be Fig. 2c
  - "leaded to faster learning" -> lead

---

> ### Author Rebuttal · Authors · 2025-07-31
>
> We thank the reviewer for their detailed comments. We are glad they found the idea of the paper "well-motivated" and "addressing an important problem", while providing "relevant" theory and a "compelling interpretation" for the mixture objective. **We will follow the reviewer's suggestions to improve the clarity of Section 5, including the motivation behind the trust-region in TRPE, which we discuss below together with all of the other raised points**. We hope our replies will make them appreciate our work even more and raise their score accordingly.
>
> **Question 1** (difference between regularized MDPs and convex Markov games)
>
> **In the Regularized MDP, the regularization is typically a function of the policy [1] and not of the state distribution as in convex Markov games**. The latter is much harder to tackle and does not allow for soft Bellman operators (see answer to Question 4 below). Moreover, the regularized MDP is typically studied in single-agent settings.
>
>
> **Question 2** (is the mixture objective novel?)
>
> Kolchinsky and Tracey studied the theoretical properties of the entropy of (general) mixture distributions and how to estimate it in general settings. As far as we know, **this work is the first considering a mixture of state distributions for unsupervised pre-training in (MA)RL**.
>
> **Question 3** (why using a trust-region?)
>
> In our setting, **a small change into the policy parameters of each agent may drastically change the value of the objective function**, i.e., the optimization landscape is often brittle. The use of the trust region, like in TRPE, allows for accounting for this effect (previous works have connected the trust region with the natural gradient [2]). While the TRPE algorithm is new, the benefits of trust-region methods in multi-agent settings have been extensively demonstrated in previous works [3]. We will make those arguments explicit in the text of Section 5.
>
>
> **Question 4** (soft Bellman operators)
>
> Even in single-agent settings, the state entropy objective can be formulated as a non-Markovian reward, as the ''value'' of being in a state depends on the states visited before and after that state. This is why **there is no hope of deriving a Bellman operator of any kind**. Since our problem is a generalization of the single-agent setting, a Bellman operator cannot be derived either. This argument is known in the literature (e.g., see [4] on convex MDPs and [5, 6] on non-Markovian rewards). Thank you for giving us the opportunity to clarify this point.
>
> **Question 5** (Figure 2 (right))
>
> **Fig. 2 (right) provides heatmaps to visualize the empirical state distributions of the two agents in the secret room Env. (i)**. Then, the clustering behavior in this context consists of the two agents focusing on disjoint portions of the state space and therefore overall exploration of the whole space.
>
> **Questions 6** (why post-learning is not better)
>
> Figure 3(c) refers to a hard exploration task with worst-case goals over the boundaries of the area reachable by the MaMujoco Reacher arm. **Random initialization may result in joint policies that never visit the goal state, and therefore, there is no learning signal at all**. The pre-trained policy is instead visiting the goal often, even without further training.
>
> **References**:
>
> [1] Geist et al. A Theory of Regularized Markov Decision Processes. ICML 2019.
>
> [2] Pajarinen et al. Compatible natural gradient policy search. Machine Learning 2019.
>
> [3] Yu et al. The Surprising Effectiveness of PPO in Cooperative, Multi-Agent Games. NeurIPS 2022.
>
> [4] Zhang et al. Variational Policy Gradient Method for Reinforcement Learning with General Utilities. NeurIPS 2020.
>
> [5] Takacs. Non-Markovian processes. Stochastic Process 1966.
>
> [6] Whitehead and Lin. Reinforcement learning of non-Markov decision processes. Artificial Intelligence 1995.

---

> > ### Comment · Reviewer_xd97 · 2025-08-01
> >
> > Thank you for the clarifications and rebuttal. I better understand the motivation behind TRPE, the novelty of the method and the experimental results. It would be beneficial to add some of what was discussed to the paper for future readers.
> >
> > My only remaining point is that it seems like it is still possible to leverage the convex MDP setting to get a state-distribution-based entropy [1]. Would this address the non-Markovian concerns (though [1] still only looks at the single agent setting)?
> >
> > [1] Brekelmans, Rob, et al. "Your policy regularizer is secretly an adversary." arXiv preprint arXiv:2203.12592 (2022).

---

> > > ### Author Response · Authors · 2025-08-04
> > >
> > > We thank the reviewer for following up on our rebuttal. We are pleased to hear most of their concerns have been addressed. We are happy to add part of the discussion to improve the clarity of the paper.
> > >
> > > Regarding the last remaining concern.
> > >
> > > If the reviewer is asking whether there is a connection between convex Markov games and state-entropy regularized MDPs: Yes, our infinite-trials formulation* can be seen as a multi-agent generalization of the state-entropy regularized MDPs without rewards.
> > >
> > > Does this address the non-Markovian concerns? No. To the best of our knowledge, there are two ways to address the non-Markovianity of the objective:
> > > - In infinite-trials formulation, one can make the reward Markovian by conditioning with the current policy. However, the contraction argument does not appear to hold for a Bellman operator over this kind of policy-based rewards;
> > > - In finite-trials formulation, one can extend the state representation to include the whole history, and then define the Bellman operator on the extended MDP. Unfortunately, this is intractable as the size of the extended MDP grows exponentially. Indeed, the optimization of the finite-trial formulation is NP-hard [1].
> > >
> > > *The finite-trial formulation, which is the target of our TPRE algorithm, is not studied in the literature of regularized MDPs.
> > >
> > > [1] Mutti et al. Convex reinforcement learning in finite-trials. 2023

---

> > > > ### Comment · Reviewer_xd97 · 2025-08-06
> > > >
> > > > I thank the author for their further clarification. Given that my concerns have been largely addressed, I have increased my rating.

---

### Official Review · Reviewer_xUL7 · 2025-06-30

**Clarity:** 3
**Significance:** 3
**Originality:** 3
**Rating:** 4
**Confidence:** 4

**Summary:**

This paper introduces a principled framework for unsupervised pre-training in multi-agent reinforcement learning (MARL) by maximizing the entropy of the state distribution—an approach well-studied in single-agent RL but still poorly understood in multi-agent contexts. The authors systematically analyze different problem formulations for unsupervised MARL—namely, maximizing the joint, disjoint, or mixture entropy of agents’ state distributions—and demonstrate that these alternatives have significant theoretical and practical differences, particularly in terms of scalability, convergence, and coordination. They propose a decentralized algorithm, Trust Region Pure Exploration (TRPE), designed to maximize state entropy in practical (finite-trial) MARL scenarios. Empirical results show that optimizing for mixture entropy strikes a favorable balance between tractability and coordination, yielding better sample efficiency and superior zero-shot transfer to sparse-reward downstream tasks, compared to joint or disjoint objectives. The work advances understanding of how to design unsupervised MARL pre-training objectives and algorithms, with mixture entropy emerging as the most effective approach for promoting coordinated exploration and efficient policy initialization for downstream multi-agent problems.

**Questions:**

1. You have mentioned the convex nature of the problem hinders the applicability of Bellman operators. Could you show me some more details about this point?
2. The error bound of the mismatch between the ideal infinite-trial and practical finite-trial cases for the mixture entropy is interesting. Let me imagine a case where there are sufficiently many agents which can almost occupy the state space. In this situation, are explorations under the joint and mixture entropies align in practical effectiveness? However, according to Theorem 4.2 the joint entropy will still lead to some errors with finite trials, while the mismatch error of the mixture entropy will be vanished. Could you show me some reasons to explain this case?
3. On line 275, what is the specification of the function $\mathcal{L}^{i}(\pi^{i}\_{\tilde{\theta}^{i}}, \pi\_{\theta^{-i}}^{-i} / \pi_{\theta})$? If I am not careless, I haven't found it across the whole paper.
4. What does the post-learning indicate in Figure 3(c)?
5. I have some question about the experimental setting of zero-shot situation. In my understanding, the goal of this task is to reach a position, though this position is randomised between four options. It seems like it still tests an algorithm's capability of exploring the state space as broadly as possible, since no algorithm has learned the knowledge of the newly set goals. As a result, it is not surprising that the ranking among the mixture, joint and disjoint in Figure 3(c) is almost the same as that in Figure 3(a) in general. Could you explain its meaningfulness, though I don't think it will affect my judgement on this paper?

**Ethical Concerns:**

["NO or VERY MINOR ethics concerns only"]

**Final Justification:**

This paper has made a solid contribution in analyzing how different types of entropy maximization make various influences to the multi-agent reinforcement learning, as a pre-training scheme. Due to the limitation of the experiments to demonstrate its potential to the large-scale cases, I recommend to retain my score as 4, but I believe this paper is worth published on NeurIPS.

During the rebuttal process, the authors have addressed my concerns in:
1. why the convex nature can hinder the use of Bellman operator
2. the influence of the thereoms in a scenario with many agents
3. the clarification of experimental settings

**Limitations:**

Yes.

**Quality:**

3

**Strengths And Weaknesses:**

### Strengths
**Quality**: This paper is well structured and with strong motivations to drive the theory of unsupervised multi-agent pre-training. This paper is complete, including results of both ideal cases such as infinite-trial objectives and practical cases such as finite-trial cases. Especially, the result in Theorem 4.2 is inspirational, underscoring the strength of the proposed mixture entropy, which can be controlled by the number of agents. The theoretical conclusion has been well incorporated into TRPO to realize off-line learning (also off-policy) by the trick of importance sampling, which makes the theoretical result able to be verified. The empirical results reflect the correctness of theoretical results.

**Significance**: The topic of pre-training is meaningful to multi-agent reinforcement learning. This paper has discussed the two common paradigms of exploration, adding one extra proposed paradigm. I believe the result can directly lead to potential applications and designs. On the other hand, the theoretical framework articulated in the paper is general enough, which I believe can be extended to other theoretical framings.

**Clarity**: This paper is written in a logical way, with sharp connection to the existing work. Each section is under guidance of a research question, which makes the whole paper always follow a story line. Most of the experimental analysis is clear and reasonable.

**Originality**: The theoretical result of this paper is original to my best experience, which directly extends the theory of convex Markov games, combined with clear research objectives, to fulfil a new theory for dealing with unsupervised multi-agent reinforcement learning.

### Weaknesses
**Clarity**: Although it can be observed that the authors have tried their best to simplify the symbols in presentation, there are still some confusing symbols.

**Quality**: The experiments are too simple to a general paper in NeurIPS. However, since the main purpose of experiments is to verify the theoretical claims, this point can be waived from my own perspective.

---

> ### Author Rebuttal · Authors · 2025-07-31
>
> We thank the reviewer for their thoughtful comments. We are especially pleased they found the topic "meaningful", the paper "well structured" and "complete", with "strong motivations", "original" and "inspirational" results, while the empirical analysis "reflects the correctness of the theory" and connection with existing work is "sharp". We provide extensive answers to their questions below in the hope they will further appreciate the paper.
>
> **Question 1**
>
> Even in single-agent settings, the state entropy objective can be formulated as a non-Markovian reward, as the ''*value*'' of being in a state depends on the states visited before and after that state. This is why **there is no hope of deriving a Bellman operator of any kind**. Since our problem is a generalization of the single-agent setting, a Bellman operator cannot be derived either. This argument is known in the literature (e.g., see [1] on convex MDPs and [2, 3] on non-Markovian rewards). We will provide those pointers in the text together with a clear explanation. Thank you for giving us the opportunity to clarify this point.
>
> **Question 2**
>
> With many agents, the behavior resulting from the two objectives might become more similar, and we believe the mean-field game regime of $ \mathcal N \to \infty $ could be an extremely interesting direction to study in future works. Yet, the upper bound in Thm. 4.2 would not be affected by having many agents. The support of the joint entropy grows exponentially with the number of agents $|S|^{|\mathcal{N}|}$ and, as such, the more agents, the more states to cover. This can generate a significant mismatch between finite-trial and infinite-trial entropy.
>
>
> **Question 3**
>
> More precisely,
> $\mathcal{L}^i (\tilde \theta^i / \theta^i) := \mathcal{L}^i (\pi^i\_{\tilde{\theta}^i}, \pi^{-i}\_{ \theta^{-i}} / \pi_\theta) = \mathbb{E}\_{d_1 \sim p_1^{\pi\_\theta} }\rho^i\_{\pi^i\_{\tilde \theta^i}, \pi^{-i}\_{ \theta^{-i}} / \pi^{i}\_{ \theta^{i}}, \pi^{-i}\_{ \theta^{-i}}} \mathcal{F} (d_1)$
> with
> $ \rho^i\_{\pi^i\_{\tilde \theta^i}, \pi^{-i}\_{ \theta^{-i}} / \pi^{i}_{ \theta^{i}}, \pi^{-i}\_{ \theta^{-i}}} = p_1^{\pi^i\_{\tilde \theta^i}, \pi^{-i}\_{ \theta^{-i}}}/ p_1^{\pi^{i}\_{ \theta^{i}}, \pi^{-i}\_{ \theta^{-i}}} = \rho^i\_{\pi^i\_{\tilde \theta^i} / \pi^i\_{ \theta^i}}.$
>
> We will include this definition in the manuscript. Thank you for giving us the opportunity to clarify this point as well.
>
>
> **Question 4 and 5**
>
> In Figure 3(c), which **refers to the Reacher environment Env (ii)**, we compare the **zero-shot performance** of different pre-trained (and random) policies against the final policy obtained by running MA-TRPO with a randomly initialized policy, hence named "post-learning". (If the Reviewer still finds this terminology misleading, we are more than happy to change it.)  In Figures 3(a,b), which **refer to the Secret Room environment Env (i)**, we compare the effect of pre-training over **the learning dynamics** by running MA-TRPO with different pre-trained policies. To extract a plot similar to Fig. 3(c) for the latter experiment, one would take the initial performance of the curves in Fig. 3(a,b) and, for the "post-learning", the final performance of the "Random initialization" curve. What they would find is that mixture entropy outperforms all of the other zero-shot average returns, and it is similar to the post-learning performance.
>
> **References**:
>
> [1] Zhang et al. Variational Policy Gradient Method for Reinforcement Learning with General Utilities. NeurIPS 2020.
>
> [2] Takacs. Non-Markovian processes. Stochastic Process1966.
>
> [3] Whitehead and Lin. Reinforcement learning of non-Markov decision processes. Artificial Intelligence 1995.

---

> ### Comment · Reviewer_xUL7 · 2025-08-01
>
> Thank you for your dedicated rebuttal. My concerns have been addressed.
>
> I suggest you may add clarifications in Question 1 (the applicability of Bellman operator), Question 3 (the notation) and Question 4 (the definition of post-learning) in the main paper.
>
> Due to that I believe the existing score can well weigh the contribution of this paper (e.g. lacking large-scale experiments), I would retain the score, but strongly recommend to get this paper accepted.

---

### Official Review · Reviewer_TW4V · 2025-07-02

**Clarity:** 2
**Significance:** 2
**Originality:** 3
**Rating:** 4
**Confidence:** 4

**Summary:**

This work explores unsupervised pretraining in multi-agent reinforcement learning via state entropy maximization. It introduces a novel mixture objective, which reduces joint state space dimensionality compared to traditional joint and disjoint objectives. The paper also provides theoretical analysis and presents a preliminary algorithm, TRPE, based on the proposed objective.

**Questions:**

1. In Line 118, is there a typo in the definition of $\xi_K$? Should it be $F_{d_K}$ instead?

2. There are some concerns regarding whether the proposed mixture objective is well-founded compared to the joint and disjoint objectives:

  1). First, does maximizing the mixture entropy necessarily lead to a well-trained joint policy?
For example, consider two agents, $m_1$ and $m_2$, with identical action spaces ${a_1, a_2}$ and identical transitions—i.e., taking action $a_1$ leads to $s_1$, and $a_2$ leads to $s_2$. A policy such that $\pi_1(\cdot|s) = a_1$ and $\pi_2(\cdot|s) = a_2$ would maximize the mixture entropy. However, this joint policy $\pi = {\pi_1, \pi_2}$ is clearly not a well-pretrained policy.
Does such a scenario occur frequently, especially in large models or with a large number of agents?

  2). From the discussion around Line 175, it seems that the mixture entropy favors agents having completely different policies over the same part of the state space. Is such behavior desirable or reasonable from a learning perspective?

  3). How can clustering, as mentioned around Line 179, be avoided? Would it be effective to add constraints or regularization terms to address this?

3. There are also some concerns regarding the simulation and result comparisons between joint, disjoint, and mixture entropy/objectives:

 1). In Line 329, the paper states that “joint and disjoint objectives fail to address mixture entropy.” This raises a question: is mixture entropy itself a reasonable metric for evaluating policy performance?
While mixture entropy may serve as a useful surrogate objective during training, it is not necessarily a direct measure of policy quality. From an information-theoretic perspective, could the paper elaborate on the validity of using mixture entropy for performance evaluation?

  2). In Figure 2 and the results on Page 22, since the algorithm is explicitly designed to maximize mixture entropy, it is expected to outperform others in terms of that metric. However, mixture entropy is not a widely accepted evaluation criterion.

As discussed in Line 73, the mixture objective is presented as a trade-off between tractability and performance. Compared with joint entropy, is it feasible to simplify the objective further, for example, by using the term $\frac{1}{N} \sum_i H(d_i^\pi)$ as a new entropy measure?

**Ethical Concerns:**

["NO or VERY MINOR ethics concerns only"]

**Final Justification:**

It addresses my concern regarding the theoretical grounding of mixture entropy and its potential to yield better performance when tasks are similar in a multi-agent reinforcement learning (MARL) environment.

**Limitations:**

Yes, the authors adequately addressed the limitations and potential negative societal impact of their work.

**Paper Formatting Concerns:**

Nan

**Quality:**

3

**Strengths And Weaknesses:**

***Strengths***
1. The paper is well written and easy to understand.
2. It proposes a novel objective—the mixture objective—which has practical significance by reducing the dimensionality of the joint state space.


***Weaknesses***
1.  I have concerns about whether the proposed objective is well-founded, specifically, whether maximizing the mixture objective truly indicates that the policies are well trained (i.e., whether it ensures sufficient exploration). I outline related questions in the Questions section.
2. The proposed algorithm, TRPE, appears to have limited applicability both theoretically and practically.

***Overall***
My primary concern is with Weakness 1. If this issue is addressed, I will consider increasing my score.

---

> ### Author Rebuttal · Authors · 2025-07-31
>
> We thank the reviewer for their insightful feedback, and we are pleased they found the paper "well written" and appreciated the "novelty" of the proposed objective. Moreover, we thank the reviewer for raising a central point about whether mixture formulations are well-founded. **We believe this concern is behind the negative score of the reviewer and we therefore provide a separate clarification** before addressing questions and concerns in detail. We hope that our replies will make the reviewer better appreciate our paper and increase their score accordingly.
>
> **General clarification: Is the mixture entropy objective well-founded?**
>
> Let us first recall that the underlying goal of unsupervised (MA)RL is to learn a general exploration policy for future tasks. Without knowledge of the tasks we will face, there is no obvious way on how to achieve this ideal objective. In the single-agent literature [e.g., 1, 2, 3], state coverage has been proposed as a valid proxy, motivating the state entropy objective. In our paper, we propose three surrogate objectives (joint, mixture, disjoint) extending the state entropy idea from MDPs to Markov games, by highlighting relative strengths and weaknesses of the formulations (see also the recap of the comparison between joint and mixture in the *"Clarification on the ideal objective"* to Reviewer Nmc9).
>
> Now, on the **mixture objective** specifically. Although we do not claim that the mixture objective is the one-fits-all solution, **why is it well-founded?**
> Whenever the task reward we will face in the (supervised) learning stage is equivalent for every agent, it does not matter which agent is visiting which state, as long as most states are covered so that we understand where the reward is. We believe this is the case in relevant practical settings (see the example below), and we designed our experiments accordingly. In those cases, the policy trained with the mixture entropy is great: The agents coordinate on visiting different portions of the state space and maximize coverage.
>
> A **practical example** in which mixture entropy is desirable: We want to train a team of "search and rescue" agents. In a specific building (environment) the target may be found in different place (different rewards) and we want to prepare for all. Mixture entropy is a good surrogate objective in this case, as the agents will split up into different portions of the buildings to traverse in order to find the target quickly.
>
> Nonetheless, there might be settings in which we aim to visit every joint state (for two agents, the reward of $r(s, s')$ is different from $r(s',s)$, i.e., the order matters). In those, the joint entropy objective is preferable, although it may be hard to estimate and optimize. At least, we know that the mixture entropy is also a lower bound to the joint entropy with a $\log (|\mathcal{N}|)$ approximation (see **Lem. 4.1**) and thus a valid proxy also in the latter case.
>
>
> **Question 1**
>
> We thank the Reviewer for spotting the typo in Line 118: $d_k$ should be $d_K$.
>
> **Question 2.1**
>
> If we understand the example correctly, we note that **the described joint policy actually maximizes the state coverage** and therefore would work well in settings where the task rewards are identical for all the agents (one of the two agents will surely get reward). This argument does not change when the number of agents increases. Instead, we agree that if the task rewards are not identical and require a specific agent to be in a specific state, then joint entropy may be better suited (although not necessarily easy to optimize).
>
> **Question 2.2**
>
> The discussion around Line 175 describes how mixture entropy favors agents having **different state distributions**. The goal is not to have "different policies over the same part of the state space" but to visit different parts of the state space. We demonstrated empirically that indeed this behavior is desirable in some settings (see also the example above for a motivating example).
>
> **Question  2.3**
>
> In many settings (see the discussion above and an example of practical applications) **clustering is exactly what one wants** to achieve.
>
> **Question 3.1**
>
> The short answer is no, **the mixture entropy is not a good metric for unsupervised pre-training in a vacuum**, as well as joint and disjoint, which are all surrogates. The real metric we care about is the average return in Figure 3, while those of Figure 2 (joint, mixture, visitation heatmaps) are useful to understand policy behaviors and the training process.
>
>
> **Question 3.2**
>
>  Let us note that **there is no widely accepted metric for unsupervised MARL**, as this is the first work addressing the problem. We propose three potential objectives, explaining their pros and cons (see also the general clarification above).
> We are happy to include analyses of further objective functions, as the one proposed by the reviewer. The proposed variation  $\sum_i H(d_i) /N$ would lead to each agent trying to maximize *their own* state entropy, without coordinating with others. This can be detrimental in some domains (see the practical example above).
>
> **References**:
>
> [1] Hazan et al., Provably efficient maximum entropy exploration. ICML 2019.
>
> [2] Mutti and Restelli. An intrinsically-motivated approach for learning highly exploring and fast mixing policies. AAAI 2020.
>
> [3] Liu and Abbeel. Behavior from the void: Unsupervised active pre-training. NeurIPS 2021.

---

> > ### Comment · Reviewer_TW4V · 2025-08-06
> >
> > Thank you for the response. It addresses my concern regarding the theoretical grounding of mixture entropy and its potential to yield better performance when tasks are similar in a multi-agent reinforcement learning (MARL) environment. I will increase my rating.

---

### Official Review · Reviewer_NmC9 · 2025-07-04

**Clarity:** 3
**Significance:** 3
**Originality:** 3
**Rating:** 5
**Confidence:** 4

**Summary:**

This paper considers the problem of unsupervised learning in multi-agent environments with centralized learning where coordinated exploration is necessary. Analogously to the use of entropy maximization in single agent unsupervised RL, the authors propose to focus on maximizing a convex combination of the individual entropies i.e. the mixture entropy. The authors then propose an algorithm based of TRPO called Trust Region Pure Exploration for achieving unsupervised MARL. On MARL domains for testing coordinated exploration, the authors find their approach also optimizes for the joint entropy as well as the mixture entropy.

**Questions:**

- Looking at Figure 2, I am a bit confused about what the ideal objective is for agents in this setting. Is the goal to maximize for the joint entropy and the authors are proposing that the mixture entropy provides a less sparse signal, which counterintuitively leads to better optimization of this objective than direct optimization? Or rather are the authors suggesting that there is some perceived value to optimizing the mixture entropy itself independently of optimizing the joint entropy?
- How were the specific policies in the right part of Figure 2 chosen i.e. after how many steps of training?

**Ethical Concerns:**

["NO or VERY MINOR ethics concerns only"]

**Final Justification:**

**Update After Rebuttal:**
- The author's addressed my confusion about the differences between maximizing for joint entropy and mixture entropy. I have raised the overall score of my review accordingly.

**Limitations:**

The theory is limited in that it does not explain the advantage of optimizing for the mixture entropy rather than the joint entropy in a formal manner.

**Paper Formatting Concerns:**

I did not notice any formatting concerns.

**Quality:**

3

**Strengths And Weaknesses:**

**Strengths:**
- This paper considers an interesting and important topic.
- Optimizing for the joint entropy would be the obvious solution for this formulation, so the proposal of optimizing for the mixture entropy is a bit unexpected.
- It is nice that the authors can provide some degree of theoretical analysis in a mostly empirical research area.
- I found the results in Figure 2 to be interesting and compelling.

**Weaknesses:**
- It is not clear to me what the theoretical value is of the mixture entropy objective rather than the joint entropy objective, the authors mention in the caption of Figure 1 that the joint entropy is sparser, implying that maybe the joint entropy is what we actually care about but it is easier to optimize for it with the mixture entropy. However, this does not seem supported based on the slow convergence in Figure 2a.
- I think the domains considered were reasonable, but the breadth of domains is not that convincing when considering that there is some confusion about what the best objective to optimize for is theoretically. As such, there may be a requirement for a high amount of empirical evidence to really have confidence about an approach in this setting.

---

> ### Author Rebuttal · Authors · 2025-07-31
>
> We thank the reviewer for their insightful comments. We are pleased they found the topic we covered to be "interesting and important", our solution "unexpected", with a "nice" theoretical analysis and "interesting and compelling" results. Moreover, we thank the reviewer for pointing out an important source of **confusion about what the intended learning objective** is and the strengths/weaknesses of joint and mixture formulations. **We believe this confusion is the reason behind the borderline score of the reviewer, and we therefore provide a separate clarification** before addressing questions in detail. We hope our replies will make the reviewer further appreciate our paper and increase their score accordingly.
>
> **Clarification on the ideal objective, joint and mixture entropy comparison**
>
> Just like in single-agent settings, the goal of unsupervised RL is to learn exploration for any possible task while interacting with the reward-free environment. If we assume the tasks are represented through state-based reward functions, the latter translates into state coverage. The state entropy is a proxy for state coverage (the argument may look convoluted, but it is common in the single-agent literature [e.g., 1, 2, 3]).
>
> Our work explores avenues to extend the state entropy objective from MDPs to Markov games, while **the ideal learning objective is still to pre-train an exploration policy for any task**.
>
> We believe the most natural state entropy formulation in Markov games is the **joint state entropy**. However, it comes with some important drawbacks:
>    - **Estimation.** The support of the entropy grows exponentially with the number of agents $|S|^{|\mathcal{N}|}$, so does the complexity of the entropy estimation problem [4];
>    - **Concentration.** The empirical entropy concentrates as $\sqrt{K^{-1}}$ for $K$ trajectories (see **Th. 4.2**);
>   - **Redundancy.** When Asm. 3.1 holds and the state space $|\mathcal{S}|$ is the same for every agent, the joint entropy may inflate state coverage as $(s,s')$ and $(s',s)$ are different joint states.
>
> In other words, the problem of optimizing the joint entropy suffers from the *curse of dimensionality*, which is particularly relevant in practice (while their difference might not be so relevant in ideal settings, see **Fact 4.1**).
>
> Another potential formulation is the **mixture state entropy**, which has the following properties:
>    - **Estimation.** The support of the entropy and therefore the estimation complexity do not grow with the number of agents;
>    - **Concentration.** The empirical entropy concentrates as $\sqrt{(K|\mathcal{N}|)^{-1}}$ for $K$ trajectories and $|\mathcal{N}|$ agents (see **Th. 4.2**);
>    - **Redundancy.** For the mixture entropy objective, the joint states $(s,s')$ and $(s',s)$ are contributing in the same way; therefore, there is no difference in visiting one or the other.
>
>
> The latter can be a limitation when we aim to explore all the possible *joint states*, e.g., when the reward functions of the agents will be different in the eventual tasks.
> At least, the mixture entropy is also a lower bound to the joint entropy objective with a $\log (|\mathcal{N}|)$ approximation (see **Lem. 4.1**) and thus a valid proxy also in the latter case, given the favorable estimation and concentration properties.
>
>
> **Question 1**
>
> As clarified above, the ultimate goal of unsupervised (MA)RL is to provide exploration for (MA)RL. This is why **the most important experimental metric to look at is the average return in Figure 3**, where the policy optimizing the mixture entropy fares well in comparison to others. The joint entropy, mixture entropy, and state coverage heatmaps of Figure 2 are interesting qualitative metrics, especially to understand how the unsupervised optimization process works, but they do not fully capture the ultimate goal in a vacuum.
>
> **Question 2**
>
> The extracted policies were the ones at the end of the training process. Indeed, we used this plot to highlight a crucial point: While it is true that mixture entropy optimization appears to lead to *slower* optimization of joint entropy in Figure 2(a), this is because of pathological behaviors. Joint entropy optimization exploits redundancy (as explained above), while disjoint entropy optimization exploits simple and uncoordinated solutions, as Figures 2(c) and 3(a,b) confirm.
>
> **References**:
>
> [1] Hazan et al., Provably efficient maximum entropy exploration. ICML 2019.
>
> [2] Mutti and Restelli. An intrinsically-motivated approach for learning highly exploring and fast mixing policies. AAAI 2020.
>
> [3] Liu and Abbeel. Behavior from the void: Unsupervised active pre-training. NeurIPS 2021.
>
> [4] Beirlant et al. Nonparametric entropy estimation: An overview. 2001.

---

> > ### Comment · Reviewer_NmC9 · 2025-08-04
> >
> > Thank you for your detailed response to the concerns raised in my review. This greatly clarifies for me why the authors argue for the mixture entropy rather than the joint entropy. I have updated the score of my review as a result. I think the paper will be improved by incorporating in this discussion.

---

### Note · Authors · 2025-08-12

Dear AC,

We believe all the reviews have been extremely instrumental in highlighting potential points of improvement and in letting us address them properly. Here, we summarise what we believe were the main and crucial points and whether they were addressed, ordered by **Strengths**, **Weaknesses**, and **Rebuttal Outcome**:

- **Rev. NmC9**:
   - They found the topic we covered to be "interesting and important", our solution "unexpected", with a "nice" theoretical analysis and "interesting and compelling".
   - They had some doubts about what the ideal objective would be in our setting, and what is the theoretical justification for the proposed objectives.
   - After a clarification and extensive justifications, by offering pointers to our theoretical results, the Reviewer confirmed that **we clarified their doubts** and **raised their score accordingly**.

- **Rev. TW4V**:
  - They found the paper to be "well written" and appreciated the "novelty" of the proposed objective.
  - They had some doubts about whether the proposed mixture objective was well-founded.
  - We provided the rationale on why the proposed objective is indeed well-founded and how it relates to the ideal objective. As a result, our rebuttal **addressed [their] concern regarding the theoretical grounding** and as a result **they raised their score**.

- **Rev. xUL7**:
   - They found the topic "meaningful", the paper "well structured" and "complete", with "strong motivations", "original" and "inspirational" results, while the empirical analysis "reflects the correctness of the theory".
   - They asked some clarifications about role of Bellman operators and some elucidations on one experimental result.
   - After clarifying all these points, the Reviewer confirmed that **[their] concerns have been addressed** and that they **would retain the score but strongly recommend to get this paper accepted**.

- **Rev. xd97**:
   -  They found the idea of the paper "well-motivated" and "addressing an important problem", while providing "relevant" theory and a "compelling interpretation" for the mixture objective.
   -  They asked some clarifications on the links with regularised MDPs and soft Bellman operators, our algorithmic choices and some experimental details.
   - Through our clarifications in the rebuttal, **[their] concerns have been largely addressed**, and thus they have **increased their rating**.

Best Regards,

the Authors

---

### Decision · Program_Chairs · 2025-09-17

**Decision:**

Accept (poster)

**Comment:**

This paper explores unsupervised pre-training in multi-agent reinforcement learning by adapting the single-agent method of maximizing state distribution entropy. The authors analyze various problem formulations, highlighting theoretical and practical challenges. They then introduce a scalable, decentralized algorithm and show through experiments that optimizing for mixture entropy provides the best trade-off between tractability and performance for downstream tasks.

This was a strong set of reviews. The reviewers unanimously praised the paper's core idea, recognizing the problem as important and the proposed solution as novel and well-motivated. The theoretical analysis was a particular highlight, with multiple reviewers pointing to the theorems as compelling evidence for the mixture objective's advantages in balancing tractability and encouraging coordinated exploration.

Despite the positive reception, the reviewers shared a key initial concern regarding the fundamental justification for the mixture entropy objective. One point, raised by two reviewers, was whether mixture entropy is simply a computationally convenient proxy for the more intuitive joint entropy, or if it is a valuable objective in its own right. They questioned if maximizing it would truly lead to effective exploration or if it could produce undesirable behaviors. Other minor critiques included a desire for more clarity and the view that the experiments, while sufficient to validate the theory, were somewhat simple for a top-tier publication.

The authors' rebuttal was effective. They successfully addressed the main point of confusion by clarifying the theoretical grounding and distinct value of the mixture entropy objective compared to the joint entropy. The authors also clarified specific technical questions and experimental details. In the end, all reviewers concluded that the paper's strengths outweighed any initial weaknesses, leading to a consensus for acceptance.